# Registered Report: How does art impact pain and stress? Exposure to multimodal art (Music + Visual) and music alone enhances pain tolerance more than visual art, but neither art form impacts autonomic or endocrine markers

Anna Fekete[ID][1,2‡*], Rosa M. Maidhof[ID][3,1‡*], Eva Specker[4,5], Andreas Gartus[6], Urs M. Nater[1,2], Helmut Leder[6,7]

1 Department of Clinical and Health Psychology, Faculty of Psychology, University of Vienna, Vienna, Austria, 2 University Research Platform 'The Stress of Life (SOLE)—Processes and Mechanisms Underlying Everyday Life Stress', University of Vienna, Vienna, Austria, 3 Translational Social Neuroscience, Department of Psychiatry, Psychosomatics and Psychotherapy, University of Würzburg, Würzburg, Germany, 4 Leibniz-Institut für Wissensmedien (IWM), Tübingen, Germany, 5 University of Tübingen, Tübingen, Germany, 6 Department of Cognition, Emotion, and Methods in Psychology, Faculty of Psychology, University of Vienna, Vienna, Austria, 7 Vienna Cognitive Science Hub, University of Vienna, Vienna, Austria

‡ AF and RMM contributed equally to this work and share the first authorship.
* anna.fekete@univie.ac.at (AF); Maidhof_R@ukw.de (RMM)

This is a Registered Report and may have an associated publication; please check the article page on the journal site for any related articles.

## Abstract

While the pain- and stress-reducing effects of music are well investigated, effects of visual art and the combination of both modalities (music and visual art) are much less explored. We tested the (1) pain- and (2) stress-reducing effects of a multimodal (music + visual art) aesthetic experience—expecting stronger effects than single modal aesthetic experiences (music or visual art)—and, in an exploratory manner, (3) investigated underlying mechanisms of aesthetic experience and (4) individual differences. In a repeated-measures design (music, visual art, multimodal aesthetic experience, control), 42 female participants submitted their self-selected movingly beautiful visual artworks and music pieces to the lab, where pain and stress were induced by a cold pressor test. Pain (global pain perception, pain intensity, pain affect, pain tolerance) and stress responses (subjective reports, autonomic [electro-cardiography, electrodermal activity, salivary alpha-amylase] and endocrine activity [salivary cortisol]) were measured. Individual differences of the experience, trait empathy and absorption were investigated. Exposure to multimodal art resulted in longer pain tolerance ($M = 80.19$s; $SD = 61.05$) compared to visual art ($M = 56.63$s; $SD = 47.86$), but not compared to music ($M = 81.34$s; $SD = 64.19$; $p < .001$; $\eta^2 = .039$). Other measures of pain intensity, stress intensity, and pain affect did not differ across the conditions. Exposure to all types of art distracted participants' attention from pain, prompted mind wandering, and elicited greater enjoyment than the control condition. While participants were overall more stressed during the cold pressor

**Data availability statement:** Data, study materials, and supplemental analyses for the present research openly available on the Open Science Framework (OSF) at: https://osf.io/yxgrv/.

**Funding:** This research was funded by two financial grants to Anna Fekete and to Rosa Maidhof, who both received a grant ("Förderstipendium") from the University of Vienna, Büro Studienpräses. The funders had no role in study design, data collection and analysis, decision to publish, or preparation of the manuscript. Funding for publishing was provided by the Open Access Publishing Fund of the University of Vienna.

**Competing interests:** The authors have declared that no competing interests exist.

test, no differences emerged across the four conditions ($p = 0.38$; $\eta^2 = .012$). Also, no differences were found regarding cortisol and alpha-amylase. Regarding individual differences, higher trait absorption was associated with longer pain tolerance in the multimodal condition ($b = 0.58$, $SE = 0.29$, $t(120)=2.02$, $p = .046$) and with lower pain intensity in the music-only condition ($b = −0.27$, $SE = 0.12$, $t(120)=−2.20$, $p = .030$), compared to the other conditions. In conclusion, exposure to art can influence pain; however, the underlying mechanisms require further research.

## Introduction

In the 2020's Oscar nominated documentary The Cave [1] which focuses on an underground hospital in Syria, Doctor Salim plays classical music to his patients during surgery telling them: "Don't worry dear, we don't have anaesthesia, but we have music!"

But can art reduce pain? And if so, can only music achieve this effect or also other forms of art? In this study we aim to understand if art can reduce pain (and stress) in order to assess if art can be an effective tool for pain and stress management.

Not only is pain one of the biggest global health problems, but its prevalence is continuously increasing as the population ages [2]. Today, pain serves as a leading reason for seeking medical care [3]. Because of the unpleasant sensory and emotional nature of pain, and the associated tissue damage [4], there is a great interest to reduce the noxious sensory and affective components of pain [5] with accessible tools in daily life.

Beside pain, stress is another global health problem [6]. The maladaptive responses of stress [7] result in bad mood, reduced sense of well-being, and negative health outcomes [8]. Given that chronic stress facilitates the development of morbidity [9], tools that can reduce stress in daily life are of great societal interest.

Pain and stress are closely intertwined. Pain is often associated with activity in stress-responsive systems like subjective acute stress, autonomic and endocrine activity [10], and can be influenced by these [11]. Therefore, concomitant investigation of both pain and stress is essential.

Art can be a powerful, and—in its digitized form—an accessible resource, creating a cost-effective tool to promote health in everyday life [12,13]. The health-related benefits of art have gained more and more attention in the last two decades as evidenced by the recent report of the WHO, reviewing literature from 2000 to 2019, on the role of art in improving health and wellbeing [13]. This report indicates that engaging with (e.g., arts and crafts activity) and participating in (e.g., attending cultural events) the arts—performing arts, visual arts, design, literature, cultural events, digital art—can be a useful tool for both physical and mental health through the components of aesthetic engagement, active imagination, sensory activation, cognitive, and emotional content [14,15], and in some cases social interactions, physical activity, health thematic coping, or interaction with healthcare. Through these components, the effects of appreciating an artwork provide a multifaceted tool that can be used to

target outcome variables on a psychological, physiological, social, and behavioural level. Aesthetic experience can also be associated with the experience of aesthetic chills, a feeling of shivering, coming along with emotional and psychophysiological reactions [16]. Although other components of art can be considered effective, here we focus on beauty as the most important component. Beauty is a psychological sensation perceived as a sensory impression; this sensation is pleasant, positive [17] and meaningful, often delightful or enjoyable. Psychologically, however, beauty is also a distinctly subjective, often even private feelings that reflects one's own experiences in a special way [18].

In this paper, our aims are the following: first, we investigate the multimodal effects of aesthetic experience on pain, and second, on stress. Specifically, we are interested in the effects of modality (music vs. visual art and their combination). Within the frame of these aims, we also investigate whether music can potentially influence pain and stress more than visual art. Third, in an exploratory manner, we investigate underlying mechanisms of aesthetic experience. Fourth, also in an exploratory manner, we investigate how individual differences play a role in pain and stress perception. Below, we first review research showing that art (music and visual art) can reduce pain and stress. Subsequently, we discuss how art can do this. Second, we discuss similarities between music and visual art in order to then discuss why multimodal aesthetic experiences (combining music with visual art) may be more beneficial than single modal experiences. Third, we discuss the underlying mechanisms of aesthetic experience and the relevant individual differences that might contribute to more effective pain and stress reduction to some people.

### Art, pain, and stress

**Music.** Music, which is used and investigated more often in pain management than other forms of art [19], can be effective both in the context of a multimodal pain management program [20] and when conducted alone [21]. With regard to the quality of music, the importance of beauty in music-based tools for the improvement of health has been emphasized [22,23]. However, even though different studies showed pain-reducing effects of pleasant (e.g., [24]) and preferred (e.g., [25–27]) music, the effects of explicitly beautiful music on pain have not yet been investigated in well-controlled studies. Beauty, however, is more than just pleasing harmony [23] since aesthetic experience integrates multidimensional psychological mechanisms [14]. A meta-analysis [21] reported that listening to music decreases pain in terms of psychological aspects of subjective pain sensation and emotional distress, furthermore physiological aspects like heart rate, blood pressure, and respiration rate decreased, and listening to music resulted in a smaller amount of analgesic intake.

Music is also commonly used in stress management [28,29]. Similar to pain, mere music listening can also reduce stress parameters [e.g., [30–33]. Also comparable with pain, the effects of beauty in music on stress are not yet investigated in controlled studies even though beautiful music is recommended for clinical use [22,23]. De Witte et al. [34] in their systematic review and two meta-analyses showed that listening to music influences subjective stress levels, as well as stress-responsive systems of the body like the autonomic nervous system and the hypothalamus pituitary axis. In sum, music can reduce both pain and stress on a psychological as well as a physiological level.

**Visual art.** When it comes to visual art as a tool for pain reduction, findings are mixed. Only one study found that beautiful paintings decreased pain perception through positive attentional distraction exhibited by pain-related cortical responses [35]. Other findings regarding the analgesic effects of visual art are controversial: for example, compared with self-selected music, visual art has not been found to influence pain tolerance nor perceived control [36].

The stress alleviating effects of viewing visual art are mixed [37] and were mainly investigated in museum and gallery settings. Gallery visits caused faster recovery from high stress assessed by salivary cortisol level [38], and having an aesthetic experience in a religious cultural heritage site resulted in a decrease of 60% salivary cortisol level [39] compared to the normally associated level of cortisol decrease during the circadian cycle (as importantly, these two studies [38,39] had no other control groups). Furthermore, artworks—especially figurative ones in comparison with modern art—decreased systolic blood pressure as an indicator of stress relief [40]. However, results are not consistent: recovery from stress was

found to be slower when viewing landscape paintings in comparison with viewing scrambled images as a control [41]. In addition, stress relieving effects of a single visual art episode [42] have not yet been tested in a controlled lab environment. In the lab setting, other confounding factors such as the physical context of the museum or gallery [43] or social influence [44] with its potential negative impact [45] could be avoided.

In summary, the effects of art on pain and stress are mixed. While music has been shown as an effective tool in both pain and stress management, the same cannot be said about visual art. Therefore, it is important to look at the underlying mechanisms—how can art influence pain and stress—in order to understand why music is an effective tool, as well as to investigate if visual (or other modalities of) art have the potential to be equally effective.

### How can art influence pain and stress?

The perception of both music and visual art includes aesthetic experiences that share pleasing and rewarding neural [46] and psychological mechanisms [47,48], therefore both music and visual art are expected to be beneficial regarding individuals' well-being through appreciation of beauty [49]. In the following, we outline how aesthetic experiences in music and visual art can influence pain and stress-responsive systems through both neural and psychological mechanisms (Fig 1).

**How can art influence pain?.** Listening to music during pain induction modulates activity regions of brain, brainstem, and spinal cord indicative of descending pain modulation [50]. Similarly, beautiful visual art can influence the activity of different cortical areas that activate pain-modulating structures in the brainstem [35]. These constant neural processes are associated with psychological mechanisms.

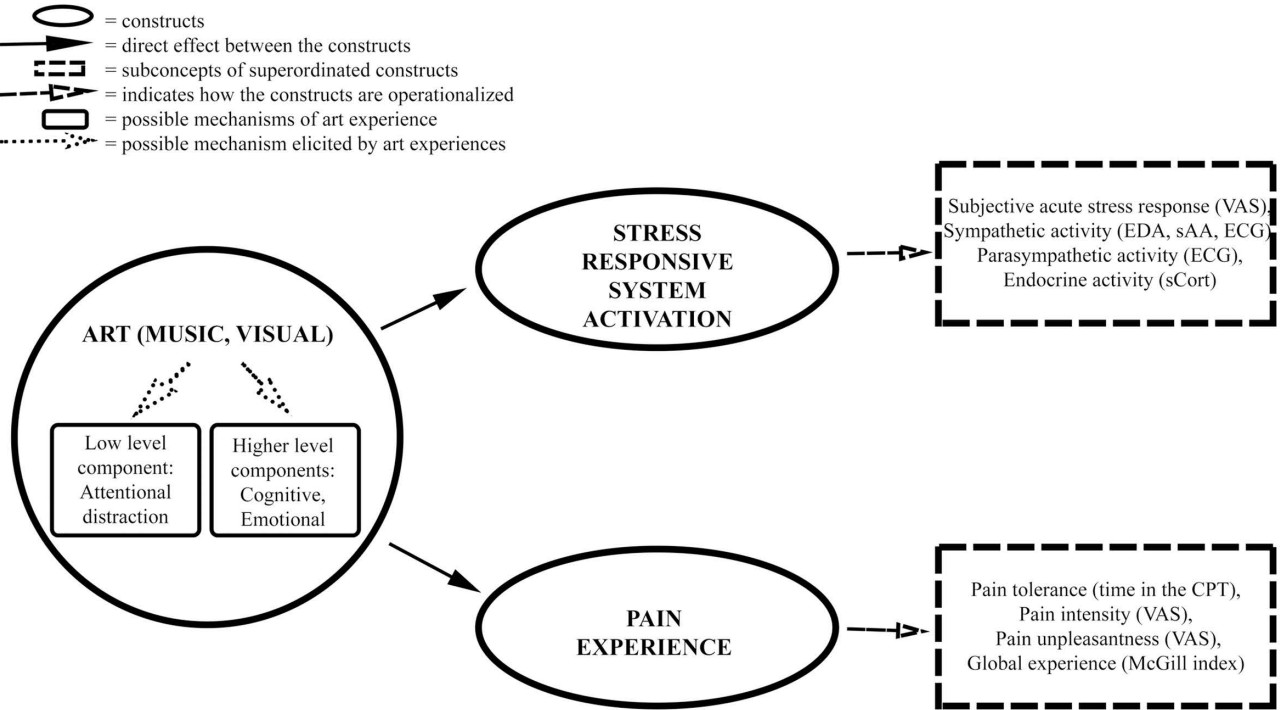

**Fig 1. Proposed mechanisms on how art can influence pain and stress.** Note. CPT = Cold Pressor Test, ECG = electrocardiogram; EDA = electrodermal activity; sAA = salivary alpha-amylase; sCort = salivary cortisol, VAS = Visual Analogue Scale.

Psychological mechanisms of pain reduction by listening to music were recently reviewed and classified into categories by Howlin and Rooney [51]. The mechanisms they propose (details below) are similar to models of visual aesthetic experience [15,42,52], encompassing first sensory-perceptual, and later cognitive and affective evaluations. Given this similarity in the underlying processes, it would theoretically be possible for visual art to be equally effective (in terms of reducing pain and stress) to music. Below, we give a detailed overview of the mechanisms proposed for music, and at each step discuss what the visual art equivalent would be.

Firstly, low level attentional resources are allocated to the music away from pain experience, which leads to a distraction from experienced pain [51]. Similarly, visual art is able to modify pain through attentional distraction [35]. Thus, both music and visual art have the potential to reduce pain by redirecting attention away to the aesthetic experience.

Secondly, higher level cognitive and emotional components have been considered as analgesic mechanisms. Through the use of music, active and focused listening, encompassing cognitive mechanisms of extra-musical associations, memories or visual imagery [53] and affective mechanisms, reduces pain [51]. Through the use of visual art, affective modulation of pain is possible [5,54,55] since aesthetic experience serves as a powerful tool to counter pain experience through the wide range of complex emotions of pleasure, contemplation, and amazement [56]. Similarly, aesthetic experience can be associated with emotional arousal, liking of the aesthetic stimulus [57] and a wandering mind [58], and these concepts can in turn be associated with pain perception [36,59,60]. Thus, both music and visual art have the potential to reduce pain by engaging thoughts and evoking emotions [61]. An important agent in this context is personal meaning which is considered as a core mechanism of pain-modulation through music listening and strengthens the sense of self [36,62]. Through listening to music, energizing, encouraging, and motivational effects result in renewed personhood that leads to physical well-being, and this effect is made stronger with self-selected music [63]. This is in line with the finding that most of the time self-selected music was found to be more effective in music interventions than pieces chosen by others [20]. Through a visual aesthetic experience, high level of self-relevance that later results in metacognitive self-awareness leads to powerful, transformative aesthetic experiences described by feeling of "cathartic" release, epiphany, enlightenment, and harmony [15,64]. Thus, both music and visual art have the potential to reduce pain more effectively through self-relevant processes.

**How can art influence stress?.** Similar mechanisms on neural and psychological levels can also explain the impact of arts on stress-responsive systems. Music can modulate activity in neuronal networks including amygdala, nucleus accumbens, and ventral tegmental area [65] that are associated with the regulation of hormonal and autonomic responses to rewarding and emotional stimuli [28]. Consistently, various studies show changes in hormonal and autonomic responses as well as subjective stress levels in response to music [30–34,65]. Similarly, it seems plausible that beautiful visual artworks can modulate stress levels since activity in the amygdala has been found for the perception of visual art [48]. Thus, both music and visual art have the potential to reduce stress by modulating attentional as well as higher order cognitive and emotional processes.

An overview of the underlying mechanisms is given in Fig 1. From our discussion, it seems evident that both music and visual art have the potential to reduce pain and stress, however, due to the similarity of the underlying processes, the question arises if a combination of music and visual art (multimodal experience) has the potential to be even more effective than music or art on its own (single modal experience). Furthermore, experiences of pain and stress are multimodal in nature. For example, pain is not only a sensory experience but also an affectively laden experience. Therefore, a multimodal tool (in our case—aesthetic experience) that aims to reduce pain and stress may be more effective than a single modal tool. We discuss this line of thinking in more detail below.

## Multimodal experience of aesthetics, pain, and stress

As noted, the experience of pain has multidimensional components on the sensory domain (intensity) and on the affective domain (unpleasantness) [5]. These are closely intertwined and cannot be selectively alternated [66], therefore tools

targeting different dimensions of pain are considered beneficial—aesthetic experience, which is associated with attentional, cognitive and emotional processes, can be such a tool.

Similarly, stress affects changes on a neural, physiological, behavioural, cognitive, and emotional level [67], again, therefore, a multifaceted approach is considered to be beneficial. These overlapping components of pain and stress can be addressed through the corresponding attentional, cognitive and emotional mechanisms of aesthetic experience.

Aesthetic experience comprising of multiple modalities—with perceptual, sensory, affective, and cognitive components [15,42,52]—is assumed to have the potential to provide a more effective tool in the context of pain and stress management than the single modalities. Multimodal aesthetic experience is assumed to be efficient through multimodal integration of attentional, cognitive and emotional aspects, and through a cross-modal transfer through the different sensory domains.

Multisensory stimulation through different senses—e.g., hearing and seeing emotional content—supports each other's effects through multimodal integration [68]. Since the literature shows increasing support for the assumption of shared attentional resources between the modalities of hearing and vision [69], it can be assumed that multimodal integration might be associated with higher attentional distraction for multimodal aesthetic experience. Also, higher level mechanisms can be assumed to be stronger activated in multimodal aesthetic experience than in single modalities. For example, targeting a wider range of emotions through music and visual art could serve as a more powerful tool than each modality on their own. In detail, multimodal aesthetic experience can be beneficial since the quality of involved emotional responses partly differs: listening to music elicits more intense vitality-related emotions like feelings of tenderness, nostalgia, peacefulness, power, joy and potential sadness [70]. In contrast, viewing paintings compared to listening to music elicits more intense wonder and reduced vitality of emotions [57,70]. Besides the quality of involved emotions, subjective complexity has a multidimensional nature regarding music and visual stimuli [71]. Its relation to hedonic value (liking, beauty, pleasantness) also differs between music and visual art: in music, the relationship between complexity and beauty followed an inverted-U curve, while in paintings, complexity was positively associated with beauty [71,72]. In summary, lower level and higher-level processes associated with music and visual art are assumed to support each other through multimodal integration of aesthetic experience.

Besides multimodal integration, multimodal aesthetic experience is also assumed to be beneficial through cross-modal transfer of different sensory domains: arousal induced by listening to Romantic piano music altered the arousal felt when looking at pictures, but the music did not transfer pleasantness [73]. On the other hand, listening to music can promote beauty judgements as it has been found that listening to nineteenth-century music resulted in evaluating faces more attractive due to misattribution of arousal [74]. As the two domains of music and visual art can alter each other, investigating aesthetic experience across various sensory domains is crucial [75]. Regarding the interplay of music and visual art, aesthetic experiences with different modalities of films, music, and visual arts were found to be described with general terms of beautiful, wonderful, and original, even though more emotionally loaded words were used for music and films than visual art [76]. In addition, aesthetic appreciation was found to be higher when people were exposed to congruent music and visual artworks: they enjoyed abstract paintings of Kandinsky more when listening to abstract music of Anton Webern and enjoyed impressionist paintings of Monet more when listening to impressionist music of Ravel [77]. Similarly, aesthetic experience was enhanced for figurative art with classical music listening, meanwhile abstract art was enjoyed more with jazz music [78]. More importantly, Actis-Grosso et al. [78] showed that paintings are experienced as more pleasant in multimodal conditions (i.e., with music) in comparison to silence conditions.

To conclude, a combination of different modalities of aesthetic experience might be an enhanced stimulation and interplay of attentional, cognitive, and emotional processes that influence the reactions from pain and stress stimuli. In order to accurately assess how aesthetic experience may reduce pain and stress, different forms of aesthetic experience need to be disentangled—studied separately and in combination. It is of high importance to further investigate modality-specific characteristics of the underlying mechanisms and to integrate them in a common framework from a multimodal perspective [79]. In order to find out if the underlying mechanisms of aesthetic experience are activated for different forms of aesthetic experience, attentional, cognitive and emotional aspects need to be investigated.

## Role of individual differences

Pain-reducing effects through music listening have been found to be stronger for persons with higher level of trait empathy, measured by the participants' tendency to transpose themselves imaginatively into the feelings and actions of fictitious characters [80]. The authors explained this by emotional engagement with the music, which is assumed to be higher for people with high levels in trait empathy, resulting in a more absorbing activity. In line with this, absorption (i.e., engagement) with music has been suggested to modulate pain experience [81]. Also, the effects of music listening on stress-responsive systems can be modulated by empathy [82]. Additionally, absorption is a promising modulator of stress-reactions to music listening [83,84]. Similar pain- and stress-reducing effects of absorption by means of visual art have not been found yet, however we argue that visual art might have this potential, too, as it has been found that when people were viewing visual art installation, their absorption increased as a function of how much awe they experience [85].

Similarly, empathy [86] and absorption [87] are also associated with emotional engagement in the context of viewing visual art. Even though, to our knowledge, the role of trait empathy and absorption has not yet been investigated in the context of pain- and stress-reducing effects of visual art, it can be assumed that enhanced emotional engagement through trait empathy and absorption would influence pain and stress-responsive systems also for the use of visual art, since underlying emotional mechanisms are assumed to be similar for the two modalities of art, as explained above.

## Present study

In the present study, we—for the first time—systematically tested: (1) the effect of multimodal (music, visual, and combined) aesthetic experience on pain, (2) stress-responsive systems; and in both cases, compared the single modalities (e.g., testing whether music or visual art is more effective); in an exploratory manner, (3) investigated the underlying mechanisms of aesthetic experience; and (4) examined the individual differences.

To our knowledge, only one study has attempted to investigate the effects of different modalities of aesthetic experience—visual art and music—on pain perception induced by a cold pressor test (CPT) [36]. However, this study has several important limitations. Firstly, although Mitchell et al. [36] included both music and visual art, they investigated only single modal experiences and did not assess the potential benefits of multimodal aesthetic experiences on pain. We aim to go further by looking at both modalities separately and in combination.

Secondly, participants could not bring their self-selected visual artwork to the lab—as they could their preferred music. In our study, we aim for consistency in the choice of the stimuli by letting participants bring both self-selected music, and self-selected visual artworks. This was necessary so that participants maintained personal meaning with their chosen stimuli—as elaborated above.

Finally, Mitchell et al. [36] only looked at pain and did not investigate the stress-responsive systems. As discussed above it is necessary to investigate pain and activity of stress-responsive systems together in order to better understand the underlying mechanisms. We address this in our study by investigating the association between pain and stress during aesthetic experience. For the investigation of the underlying mechanisms of aesthetic experience, we investigated the activation of attentional, cognitive and emotional aspects. In order to account for individual differences, the role of trait empathy and absorption was investigated.

The benefits of this research are multifaceted. For the first time, the effects of the single modalities of music and visual art on pain and stress responses are systematically compared with each other and with multimodal (music+ visual art) aesthetic experiences. By investigating the underlying mechanisms of aesthetic experience, our findings provide insight into which facets of aesthetic experience (attentional, cognitive and emotional processes) have the greatest influence on the correlates of pain and stress. Therefore, future art interventions will be able to target these most influential facets and thus become more effective. By investigating personality differences, individually tailored art interventions are possible. As our approach combines clinical psychology and empirical aesthetics, our findings contribute to the development of individualized pain and stress management programs through art modalities, thereby supporting the general improvement of well-being in society.

## Hypotheses

Detailed hypotheses for each combination of dependent and independent measures can be found in Supplementary S1 Table that provides a summary of research questions, hypotheses, analysis plan and interpretations. Broadly speaking we tested two confirmatory hypotheses:

1. We hypothesized that (1.1) multimodal (music + visual art) aesthetic experience would reduce pain more than single modal experience; and that (1.2) music would reduce pain more than visual art.

2. We hypothesized that (2.1) multimodal (music + visual art) aesthetic experience would reduce activity in stress-responsive systems more than single modal experience; and that (2.2) music would reduce activity in stress-responsive systems more than visual art.

In an exploratory manner, we investigated (3) the lower (attentional) and higher (cognitive and emotional) processes of potential underlying mechanisms on how multimodal aesthetic experience influences pain and stress experience. Furthermore, we explored (4) individual differences—specifically trait empathy and trait absorption—that might influence the effects of aesthetic experience on pain and stress.

## Materials and methods

### Design

We used a within-subjects design with four conditions: music, visual, multimodal, and control (see Fig 2 for the conditions). A within-subjects design was chosen because this was found to be the most suitable for pain research due to the wide range of significant individual differences that occur in response to pain [88]. However, this comes with the need to have sufficient length of washout period between the trials and prevent carry-over effects. To ensure that these conditions are met, each of the 4 conditions were administered during a separate session on a different day (at least 24 hours between sessions since we expect this elapsed time is sufficient to prevent carryover effects, however, this timeframe is exploratory since currently no research study is available on this regard). Thus, participants needed to come to the lab 4 times (1 condition per session) on 4 different days. The order of conditions was randomized (with random number generator) and counterbalanced. We used Latin square to counterbalance the sequences of the conditions (with 4 conditions, 4! = 24 possible sequences, therefore these 24 sequences were repeated 1.5 times since we calculated a sample of 36 participants—for the sample size calculation, see "Participants" section below). Testing time was always in the afternoon (between 11:30 am and 5:00 pm), since it has been found that pain perception was lowest during the mornings if pain is induced by cold [89] and because of the diurnal changes in cortisol [90].

### Participants

Forty-two female participants ($M_{age}$ = 21.03 years, $SD_{age}$ = 1.85, range = 18–25) took part in the study. Only female participants were included to the study as females have been shown to display higher sensitivity to stress and music compared to males [91]. All experimenters were female. This consideration was introduced to avoid social desirability and societal biases around pain tolerance in a mixed gender environment where individuals with a more masculine (and less feminine) gender identity might feel more prone to affirm their "toughness" or "bravery in the presence of an experimenter from the opposite sex" [92]. The data collection took place from 14th of November 2022–10th of November 2023.

Sample size was calculated with G*Power 3 [93] by using repeated measures ANOVA by providing four groups of participants (see hypothesis 1 for the main investigation of our primary outcome variables), hence the four measurements. Medium effect size of Cohen's $f$ of .23 was used since this effect size has been found for the effect of self-selected music stimuli on pain intensity reduction in adult sample of a *Cochrane* meta-analysis and review [94] (Note that Cepeda et al. [94] reports an overall effect size (mean difference) of −.46 that we transformed to Cohen's $f$ of .23 because that

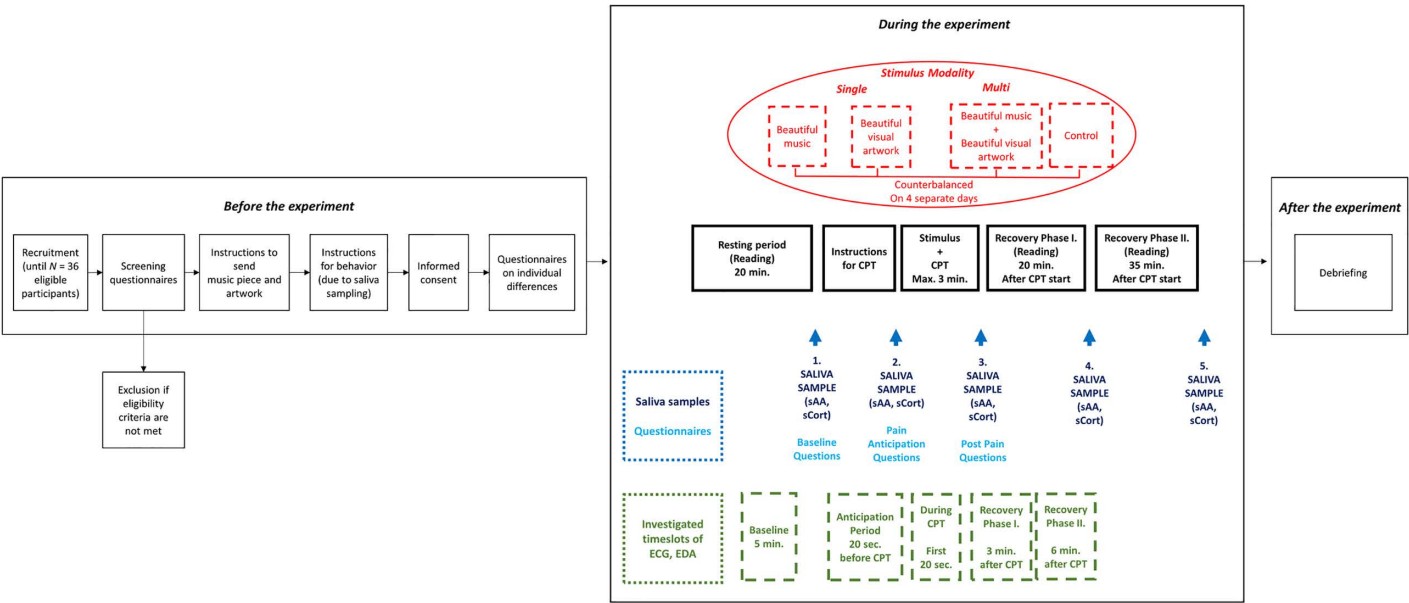

**Fig 2. Procedure of the study (Before, During, After the Experiment).** Note. CPT = Cold Pressor Test; ECG = Electrocardiogram; EDA = Electrodermal Activity; sAA = Salivary Alpha-Amylase; sCort = Salivary Cortisol.

was needed for G*Power sample size calculation). The power analysis indicated a total sample size of 36 participants to achieve a power of 0.90 with an α of .05, while the correlation among repeated measures was assumed to be of 0.5. Since drop-out rates may be high (participants have to complete 4 separate sessions), we intended to collect data until we have complete data (i.e., 4 completed sessions) for 36 participants. We sampled 42 participants to increase power up to .95. Participants were recruited through the participant database system of the University of Vienna, via flyers, and social media. The online platform contained a description of the study, informed participants about the eligibility criteria and they registered to their preferred dates and timeslots for the study. Interested persons were then contacted by the experimenter and were checked for eligibility criteria with detailed online screening questionnaires. Participants were included and excluded based on standard practice that have been used in, e.g., [30,31].

## Eligibility criteria

Participants were included if they were between the age of 18 and 29 years (Emerging Adulthood [95]), had an average Body Mass Index (BMI) from 18.5 to 25 kg/m², spoke German fluently, and had regular menstruation. Participants were excluded from the study if they had colour blindness, inappropriate visual acuity, hearing problems, art or music-related profession, or art- or music-related studies, mental disorders, cardiovascular diseases, arterial occlusive diseases, very high or very low blood pressure, chronic pain, diabetes, Raynaud's syndrome, epilepsy, or a recent serious injury. Further exclusion criteria included regular intake of pain-reducing medication, or medication intake with impact on stress-responsive systems, smoking more than five cigarettes per week and current drug consumption. Additional exclusion criteria of female participants included: pregnancy or breastfeeding, premenstrual syndrome, use of hormonal contraceptives.

We are aware that the strict eligibility criteria bring the limitation of generalization of the findings. However, because of the novelty of the study, it was necessary to make sure that we do not risk participants' safety (see also, e.g., McIntyre et al. [96]) and exclude participants with characteristics that influence the perception of the presented stimuli (see also, e.g., Madsen and Moore [97]) or our outcome variables (see also, e.g., Adam and Kumari [98]; Carr and Goudas [99].

Therefore, we only tested those persons who met these eligibility criteria, in accordance with the relevant literature (e.g., Linnemann, Ditzen, et al. [30]; Linnemann, Strahler et al. [31]).

Participants received course credit (16 credits) or monetary compensation (40€) for participation in all sessions. Partial compensation of 2 credits/5€ was granted after each of the first three sessions, and 10 credits/25€ was given after attendance of the last session. The study was approved by the ethics committee of the University of Vienna (reference number: 00506) and was carried out in accordance with the Declaration of Helsinki.

## Materials

**Stimuli.** Participants were asked to provide self-selected pieces of music and artworks that they find "movingly beautiful" (see also [100,101]) because they reliably elicit beauty and pleasure [102]. In the music and visual artwork conditions, the stimulus was presented as single modality.

Participants sent a digital reproduction of their self-selected movingly beautiful piece of music or visual artwork, via e-mail prior to the experiment. To ensure congruency, we asked participants to select music and visual art that are a good fit together.

In case of self-selected music, the pieces should not have been longer than seven minutes and no shorter than three minutes. The maximum length aims that the main content of the music piece falls into the CPT exposure (e.g., if the music piece is much longer, it is possible that the first three minutes are an introductory part and the most powerful part of the piece comes later when the CPT is over), whereas the minimum length of the music piece reflects the length of the CPT (3 minutes). The piece of music should not have contained breaks or silences, meaning that participants should not be exposed to silence any time during the CPT. The selected recording of the piece of music should have had a good sound quality. The pieces of music were displayed with active acoustic noise cancelling JBL LIVE 460NC headphones.

With the self-selected visual artworks, the images were required to have a minimum resolution of 720×576 pixels.

The visual artworks were presented with a flexible code that adjusts the presentation size depending on the resolution of the image to be presented the best way on an LCD screen of 30 inches (LG Flatron W3000H), with a capacity of 2560 x 1600 pixels resolution. This process was used to ensure perceived control and fulfil the self-selected criteria in the movingly beautiful music/visual artwork condition.

In the multimodal condition, the self-selected music piece and the self-selected artwork were displayed at the same time, again during the pain induction. In the control condition, participants were instructed to wear noise-cancelling headphones with no auditory stimuli display and view the grey screen during pain induction.

We observed that there were some artists that were more likely to be chosen by participants. Regarding visual artworks, seven persons selected artworks from Claude Monet and four of these were one of the water lilies paintings, and three persons selected artworks from Van Gogh. Regarding music pieces, four persons selected piano music, and two persons selected music pieces from the following artists respectively: Coldplay, Lana del Ray, Ludovico Einaudi, Taylor Swift, Vivaldi, Cro, and Alt-J (see the full stimuli set on OSF: https://osf.io/yxgrv).

**Cold pressor test.** We used the CPT since it was found to reliably and effectively elicit pain experience [103,104]. The CPT was found to be applicable to induce stress response [105], for example by increasing HPA axis activity, resulting in an increased cortisol concentration [106], as well as by increasing sympathetic nervous system activity, resulting in increased skin conductance [107] and elevated blood pressure [106].

Participants were asked to immerse their dominant hand—since the non-dominant hand is equipped with EDA sensors [108]—to a circulating cold-water bath and to leave it there until it is too uncomfortable to continue. They were asked to immerse the hand up to the wrist, avoid forming a fist with their fingers, and refrain from touching the container. Crushed ice was used to make sure that the water remained cold for the duration of the test, and the ice was covered with a metal grid to prevent contact. The temperature of the water was kept constant at around 1°C (acceptance range:>0.5 °C and <1.5 °C)—as suggested by Fanninger et al [109] when assessing pain tolerance—and monitored by a digital probe

thermometer (ATP Messtechnik GmbH Ettenheim, resolution = 0.1°C, accuracy = ± 0.5°C). The water temperature was on average $M = 0.65$ °C ($SD = 0.05$, Min = 0.6, Max = 0.8 °C). The water was constantly circulated with a pump to avoid laminar warming around the submerged hand. During the CPT, the stimulus of one condition was presented with the same onset of the stimuli and hand immersion. The maximum duration of the CPT was 3 minutes.

### Measures

**Pain measures.** Pain tolerance was measured in time [s] once per condition. The experimenter stood behind the participant with a portable keyboard, and when the participant submerged their hand into the cold water, the experimenter pressed the onset button which activated the visual stimulus—appearing on the screen/ the music stimulus—playing the music through headphones. When the participant removed the hand from the cold water, the experimenter pressed the button again, indicating the offset of the CPT, and the end of the stimulus presentation.

Global pain perception was measured by the McGill Pain Questionnaire [110,111] after the CPT (see also S1 and S3 Tables), furthermore pain intensity, pain affect and perceived control were measured once per condition (right after CPT, third timepoint) with Visual Analogue Scales (VAS; [112])—a continuous line on which participants rated their perception with 0 (e.g., meaning in the VAS of pain intensity: no pain at all) and 100 (e.g., meaning in the VAS of pain intensity: most intense pain that can be imagined) as end points.

Finally, momentary pain level was measured five times per conditions with the item "I feel pain" which was rated with a VAS ranging from 0 (not at all) to 100 (a lot).

**Stress measures.** For the assessment of the subjective acute stress level, i.e., momentary stress, the item "I feel stressed" was rated with a VAS ranging from 0 (not at all stressed) to 100 (maximally stressed) and was recorded five times per conditions. Furthermore, stress intensity was recorded once per condition (right after the CPT, third timepoint). The use of a single VAS for the assessment of subjective acute stress has been validated in a clinical context [113] and is also widely used in other studies in related research fields (e.g., Feneberg et al., [114]; Thoma et al. [32]).

**Physiological measures of stress:** In order to quantify the ANS activity, electrocardiogram (ECG) was used for the investigation of heart rate variability (HRV) which is quantified in terms of RMSSD (i.e., square root of the mean squared differences of successive heartbeat intervals) as an indicator for parasympathetic activity [115]. Heart rate (HR) was calculated in terms of beats per minute (BPM) as an indicator for both sympathetic and parasympathetic activity. For more information about the physiology filtering and other data preprocessing details, see Supplementary Material Section 1. Electrodermal activity (EDA) was assessed by using skin conductance level (SCL) as a parameter for sympathetic activity [116]. Room temperature was held constant between 23 and 24°C to ensure stable EDA.

ECG and EDA data was measured by 8-channel bioamplifier (Mobi 8 BP, TMS International, Enschede, the Netherlands) with a 24-bit A/D conversion rate. Physiological data was sampled at 256 Hz and continuously stored on a hard drive. The amplifier was connected to a separate computer and recorded in MATLAB 9.8 (MathWorks, Inc, USA). ECG active electrode was placed on the left collarbone, meanwhile reference and ground electrodes are placed on the left and right mastoid bones. EDA was measured by two electrodes which are placed on middle and index fingers of the non-dominant hand [108]. The acquisition of unfiltered raw skin conductance data was guaranteed using a custom-specific skin conductance sensor. Before electrode application, the skin was cleaned using curd soap. ECG and EDA parameters were constantly measured but only certain time slots were investigated (a baseline interval of 5 minutes during the initial rest period, 20 seconds immediately before pain induction to indicate anticipation to the CPT, the first 20 seconds during the CPT—those participants who attended the CPT shorter than 20 s are excluded from ECG and SCL analysis because quality of calculated physiological parameters cannot be ensured—, 3 minutes directly after the CPT as a first part of the recovery period, and the subsequent recovery period of 3 minutes).

**Salivary stress biomarkers:** Before coming to the given session, participants were asked to follow instructions in order to reduce confounding factors that can affect salivary alpha-amylase and cortisol. The instructions were sent by

email 24 hours before the session to those participants who met the inclusion criteria. These instructions were standard procedure and based on related literature [117]. Participants were instructed not come to the lab if they feel sick; not to eat in the previous one hour before the session starts; not to use chewing gum within the previous 24 hours before the session starts; not to brush their teeth in the hour before the session starts; not to smoke, not to drink caffeine-containing beverages (coffee, tea, coke), juice, alcohol in the previous 18 hours; not to engage in physical activity in the previous 24 hours; and not to drink water 10 minutes before the sampling. (If a participant did not follow one part of the instructions (we asked participants to fill out a questionnaire at each session to check if they followed the instructions), we made a note, and asked them to follow this practice for each upcoming session (e.g., if someone had a coffee in the morning before the session in the afternoon, then we asked this participant to follow this practice for the upcoming sessions in order to have the same treatment within the participant. We only excluded participants if they felt sick or we saw them drinking a coffee and eating something, because this would have meant that they did consume food/caffeine in the last hour before the saliva sampling.).

When the participant arrived, we asked them to drink a glass of water in order to remove potential aliment remains and to better standardize the mouth moisture and emphasized that they can only drink water up to 10 minutes before the sampling.

In each of the 4 sessions, five saliva samples were collected by means of SaliCaps (IBL, Hamburg, Germany), resulting in 20 samples in total per person. Saliva samples were obtained by collecting the saliva after two minutes without swallowing into a tube by means of a straw. Immediately after collection, samples were stored at −20 °C until analysis. In order to gain information on endocrine activity, salivary cortisol (sCort) was investigated, using a commercial luminescence immunosorbent assay (LUM; IBL, Hamburg, Germany). Also, salivary alpha-amylase (sAA) was investigated, an enzyme used as an indicator of the activity of the sympathetic nervous system [118]. Therefore, a kinetic colorimetric test with reagents from DiaSys Diagnostics Systems (Holzheim, Germany) was applied. Total intra- and inter-assay coefficient of variation was < 10%. For an overview of all pain and stress measures on the theoretical and measurement levels, see also S1 Fig.

*Mechanisms of aesthetic experience*: In order to get insight which mechanisms are the most influential when art influences pain and stress, we conducted measurements in all conditions: music, visual art, multimodal, control. First, we measured lower level, attentional processes, we asked participants to evaluate their level of *Distraction* that was measured with the item: "I was distracted from the cold water by the music and visual artwork/ music/ visual artwork/ the grey screen and the headphones".

Regarding the higher level, we measured *Mind wandering* "The [stimulus] made me think about other things, such as memories and emotions."

Furthermore, we measured, in an exploratory way, *Pleasantness* "I found the [stimuli] emotionally positive"; and *Emotional Arousal*: "I found the [stimuli] emotionally arousing"; *Liking* "I liked the [stimuli]."; and complex emotions: "The [stimulus] has evoked the following emotions: joy, sadness, relaxation, anger, fear, nostalgia, melancholy." *Personal meaning* was measured with "I found the [stimuli] personally relevant." Aesthetic experience as a whole was measured by *Beauty* with the following items: "I found the [stimuli] movingly beautiful in this laboratory environment"; by *Chills* (after an explanation of the term "chills"): "I experienced chills while being exposed to the [stimuli]."; and by *Enjoyment* "I enjoyed [looking at/ listening to the stimulus]." In the multimodal condition only, we measured *Congruency*: "I found that the music piece and visual artwork fit well together." All items had to be rated on a 5-point Likert scale (from "not at all" to "very much").

**Individual differences.** We assessed two underlying mechanisms of how people may be influenced by art in a pain- and stressful situation (CPT) by looking at two aspects of the experience (i.e., measures of distraction, mind wandering, as well as qualitative data of participant's motivations for choosing the art pieces) as well as aspects of the person (i.e., individual differences of trait empathy and trait absorption).

We investigated an indicator of trait empathy with the Questionnaire of Cognitive and Affective Empathy (QCAE; [119]). Items were answered on a 4-point scale ranging from 1 (= "strong disagreement") to 4 (= "strong agreement").

Trait absorption was measured the German version of the Tellegen Absorption Scale (TAS; [120]; based on the original English version by [121]. The questionnaire consists of 34 items that can be answered on a 5-point rating scale. For an overview of the assessments in the whole course of the study, see also S2 Table.

## Procedure

After the screening procedure for the testing of eligibility criteria (see above) participants were contacted to send the artwork and the music piece. Once they had sent their self-selected stimuli set, they were instructed to follow instructions regarding the saliva sampling.

Subsequently, participants came to the lab four times. Participants were asked to sign a written informed consent form in the presence of the experimenter, which stated that they could withdraw their participation and request the removal of their collected data at any time during the data collection period, without providing any reason.

Each session was conducted in a quiet lab environment, and each session followed an identical procedure (see Fig 2), except for the first session, during which participants were asked to complete questionnaires on trait empathy and trait absorption before the pain and stress exposure. Initially, participants were seated in a comfortable chair, approximately 1 m in front of the monitor, and their non-dominant hand was equipped with devices of electrocardiography (ECG) and electrodermal activity (EDA) to ensure the continuous measurement of ECG and EDA during the whole session. During the resting period of 20 minutes—because the HPA axis has a reaction time of about 20 minutes for cortisol secretion [122]—participants read natural science journals. Afterward, the first saliva sample was taken as a baseline measurement in order to investigate salivary alpha-amylase (sAA) and salivary cortisol (sCort). Then, participants completed a pre-test consisting of questionnaires on pain and subjective stress levels. Subsequently, participants were informed that a CPT would follow to induce pain and stress. Immediately before the CPT, in an anticipatory period, participants were asked to give a second saliva sample and complete questionnaires on pain perception and subjective stress levels. This was followed by the CPT at the same time with the stimulus exposure. We told participants to keep their eyes open for the whole experiment. If we noticed someone would close their eye, especially during the music only condition, we told them to open their eyes to keep this comparable across conditions. After the CPT, participants gave a third saliva sample and completed a post-test consisting of questionnaires on perceived pain, stress, and perception of the stimuli. During the subsequent recovery period, participants gave two more saliva samples (fourth saliva sample 20 minutes after CPT start, and fifth saliva samples 35 minutes after CPT start) and completed questionnaires on subjective pain and stress levels at both time points during the recovery period. Each session lasted for about 70 minutes.

## Statistical analyses

To test hypotheses 1 and 2, two-factor repeated-measures ANOVAs were conducted in R (version 4.4.1. R Core Team, 2024 [123]) by using the packages *ez* (version: 4.4.0, [124]) and *afex* (version: 1.4.1, [125]) and *BayesFactor* (version: 0.9.12.4.7, [126]). Momentary pain was investigated five times per condition. Pain tolerance, pain intensity, pain affect, and Global McGill Pain Index were investigated only once per condition, right after the CPT (see also S1 and S3 Tables). Regarding our secondary stress measures, momentary stress was investigated at five time points, similarly for physiological stress measures with the same time slots for ECG and EDA parameters (see also S1 Table). *Condition* was included as a factor with four levels (music, visual art, multimodal aesthetic experience, control). Values of $p < .05$ were considered as significant in all analyses. For more details see S1 Table. Summary of research questions, hypotheses, analysis plan and interpretation, and our OSF page: https://osf.io/yxgrv/.

## Results

### Pain and stress manipulation check

In this section, we describe the effect on time, testing if participants experienced more pain and stress in the CPT compared to baseline and recovery, accounting for the validation of our design including pain and stress manipulation. We

assessed all stress measures according to the five time points—I *Baseline*, II *Anticipation*, III *After CPT*, IV *Recovery 1*, V *Recovery 2*—of each session. For I *Baseline* we sampled 5 minutes resting time, for II *Anticipation* 20 seconds before CPT, for III *During CPT*, the first 20 seconds of CPT exposure, for IV *Recovery 1*, 3 minutes after CPT, for V *Recovery 2*, 6 minutes after CPT. (The effect of condition will follow in the "Effects of Aesthetic Experience on Pain" section.).

## Manipulation check for subjective measures

For momentary pain (VAS scores), we found a main effect of time ($F(1.11, 45.52) = 214.88$, $p < .001$, partial $\eta^2 = .84$), indicating that people experienced higher levels of pain directly after the CPT, in comparison to the baseline, anticipation, and recovery timepoints (Fig 3A, S4 Table). This validated our design by showing that our pain induction was successful.

For momentary stress (VAS scores), similar to the momentary pain, we found a main effect of time ($F(1.52, 62.47) = 40.31$, $p < .001$, partial $\eta^2 = .50$). In detail, people experienced higher level of stress after the CPT compared to the baseline, anticipation, and recovery timepoints, again, validating our design by showing that our stress induction was successful. (Fig 3B, S5 Table). This subjective stress and pain rating during CPT is in line with previous studies in the field (e.g., [127]).

In order to disentangle the effects of art encounters from perceived control—as suggested by [128]—we compared perceived control in the four conditions: visual only, control, music only, multimodal. This was relevant as we used self-selected stimulus material for the art conditions (i.e., people sent their own preferred music and visual art), while we used researcher selected material (i.e., grey screen) for the control condition. To exclude the potential that the effect is due to more perceived control over the pain and stress in the self-selected art conditions and less control in the control condition of grey screen, we asked people how much control they felt in each condition. We found that perceived control over the pain and stress did not differ between the conditions (see Table 2 for results and Table 1 for descriptives). Therefore, if the art conditions performed better than the control condition, the effect could be attributed to the influence of art rather than to differences in perceived control across the conditions.

**Manipulation check for physiological measures.** Regarding physiological measures of stress, for RMSSD, we did not find an effect of time ($F(2.26, 18.10) = 2.27$, $p = .127$, partial $\eta^2 = 0.22$; Fig 5A, S6 Table). In contrast, for HR, we found a main effect of time ($F(2.56, 46.06) = 17.12$, $p < .001$, partial $\eta^2 = 0.49$; Fig 5B, S7 Table), meaning that

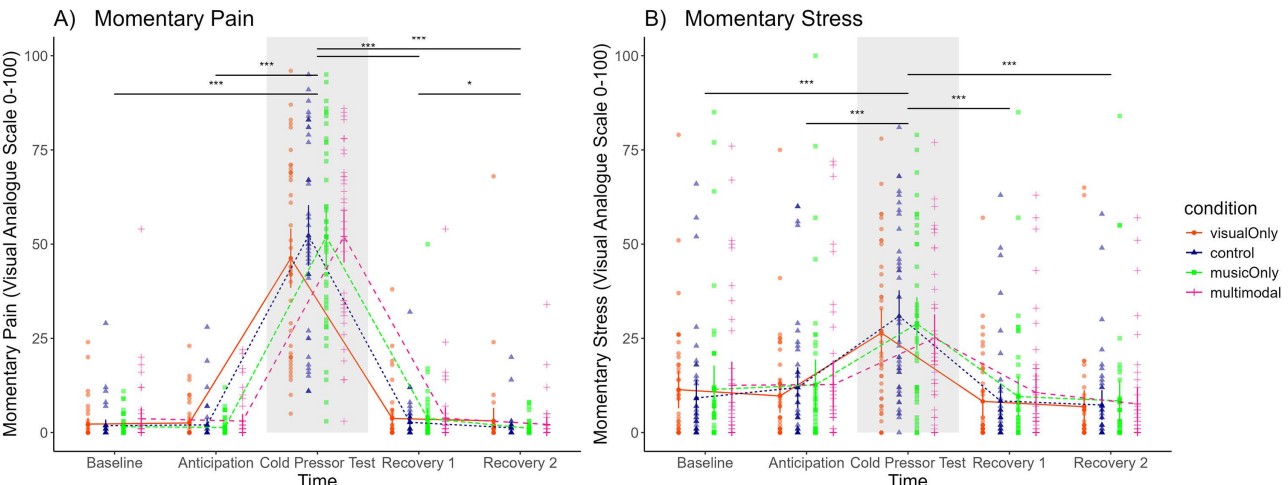

**Fig 3. Momentary Pain (A) and Stress (B) according to the five time points.** Note. A) For momentary pain, we found main effect of time, indicating that people experienced higher level of pain directly after the CPT, in comparison to the baseline, anticipation, and recovery timepoints. B) For momentary stress, we found a main effect of time, participants experienced higher level of stress after the CPT compared to the baseline, anticipation, and recovery timepoints. Significance levels:*: p ≤ 0.05; **: p ≤ 0.01; ***: p ≤ 0.001; ****: p ≤ 0.0001. N = 42.

**Table 1. Descriptive of Pain and Stress Measures.**

| Measures | Condition | | | |
|---|---|---|---|---|
| | Visual<br>M (SD) | Control<br>M (SD) | Music<br>M (SD) | Multimodal<br>M (SD) |
| Pain tolerance (in s) | 56.63 (47.86) | 59.50 (55.86) | 81.34 (64.19) | 80.19 (61.05) |
| Pain intensity (VAS) | 60.40 (22.07) | 64.90 (21.38) | 58.48 (24.21) | 58.14 (23.58) |
| Pain affect (VAS) | 64.45 (19.21) | 67.67 (21.11) | 61.88 (21.81) | 59.26 (24.47) |
| Stress intensity (VAS) | 41.83 (24.23) | 46.79 (26.27) | 45.17 (28.23) | 39.40 (25.97) |
| Perceived control (VAS) | 70.73 (23.39) | 68.88 (23.61) | 75.95 (18.52) | 72.43 (20.89) |
| McGill sensory pain index | 16.12 (6.25) | 16.24 (5.73) | 16.69 (6.54) | 15.52 (6.40) |
| McGill affective pain index | 2.14 (2.14) | 2.05 (1.92) | 1.57 (1.62) | 1.48 (1.95) |
| McGill general pain index | 3.00 (0.86) | 3.17 (0.88) | 2.95 (0.82) | 2.95 (0.85) |
| McGill total pain Index | 18.26 (7.81) | 18.29 (6.62) | 18.26 (7.05) | 17.00 (7.51) |

*Note: VAS: Visual Analogue Scale.* McGill global pain index (Short form, [110]).

**Table 2. ANOVAs of Pain and Stress Measures.**

| Measures | Effect | df | F | p | Generalized $\eta^2$ | $BF_{10}$ ± Margin of Error % |
|---|---|---|---|---|---|---|
| **Pain tolerance (in s)** | Condition | 3, 123 | 9.417 | < 0.001 *** | 0.039 | 1508.873 ± 0.49% |
| **Pain intensity (VAS)** | Condition | 3, 123 | 2.889 | 0.038 * | 0.014 | 0.920 ± 0.46% |
| **Pain affect (VAS)** | Condition | 3, 123 | 3.216 | 0.025 * | 0.021 | 1.379 ± 0.71% |
| **Stress intensity (VAS)** | Condition | 3, 123 | 1.035 | 0.379 | 0.012 | 0.106 ± 0.68% |
| **Perceived control (VAS)** | Condition | 3, 123 | 1.989 | 0.119 | 0.015 | 0.322 ± 0.58% |
| **McGill sensory pain index** | Condition | 3, 123 | 0.886 | 0.451 | 0.005 | 0.087 ± 0.52% |
| **McGill affective pain index** | Condition | 3, 123 | 3.947 | 0.009 | 0.023 | 3.203 ± 1.07% |
| **McGill general pain index** | Condition | 3, 123 | 1.508 | 0.216 | 0.011 | 0.182 ± 0.5% |
| **McGill total pain index** | Condition | 3, 123 | 1.214 | 0.308 | 0.006 | 0.127 ± 0.49% |

Note. VAS: Visual Analogue Scale.; VAS ranged from 0 to 100; BF: Bayes Factor + Percentage uncertainty; Significance levels: *: $p \le 0.05$; **: $p \le 0.01$; ***: $p \le 0.001$; ****: $p \le 0.0001$.

(Bayes Factor interpretation. $BF_{10}$ provides evidence for two competing hypotheses and shows the ratio of the likelihood of the data falls under one hypothesis to the likelihood that the data falls under the other hypothesis. $BF_{10}$ indicates the Bayes factor comparing the alternative hypothesis ($H_1$) to the null hypothesis ($H_0$). Interpretation: $BF_{10} < 1$: Evidence for the null hypothesis ($H_0$), $BF_{10}$ between 1 and 3: Anecdotal evidence for the alternative hypothesis ($H_1$), $BF_{10}$ between 3 and 10: Moderate evidence for $H_1$, $BF_{10}$ between 10 and 30: Strong evidence for $H_1$, $BF_{10}$ between 30 and 100: Very strong evidence for $H_1$, $BF_{10} > 100$: Extreme evidence for $H_1$ [130,131].

people experienced reduced heart rate from the CPT to the Recovery 1 ($p = .003$) and from the CPT to Recovery 2 ($p < .001$). Similarly, for SCL, we found a main effect of time ($F(2.05, 75.79) = 52.35$, $p < .001$, partial $\eta^2 = .59$; Fig 5C, S8 Table), indicating that SCL increased from the baseline to the anticipation phase and, as expected, increased more from the anticipation to the CPT phase due to the stressor, followed by a gradual decrease at the recovery time points.

For salivary stress biomarkers, as the data was non-normally distributed, we natural log-transformed both sCort and sAA [ln ($x$) + 10] as suggested by previous studies [117,129].

Regarding salivary alpha-amylase (sAA), repeated measures ANOVA showed that sAA changed over time ($F(3.08, 117.15) = 5.51$, $p = .001$, partial $\eta^2 = .13$). More specifically, post hoc tests using Bonferroni correction indicated that sAA increased from baseline to anticipation ($p < .001$), from baseline to the time point directly after the CPT ($p = .008$), and from after the CPT to Recovery 1 ($p = .001$). Apart from these, no further changes over time emerged ($p > .05$). These results

indicate that the CPT did indeed induce an sAA response, which did not attenuate towards the recovery phases (Fig 6A, S9 Table).

Regarding salivary cortisol (sCort), repeated measures ANOVA revealed that sCort did not change over time ($F(1.21, 45.80) = 3.04$, $p = .08$, partial $\eta^2 = .07$). This indicates that the CPT was not strong enough to induce a change in sCort (Fig 6B, S10 Table).

Exploratory linear mixed-effects models (LMMs) were conducted to examine the effects of time and condition on all dependent variables. The first model included a random intercept for participants (DV ~ timepoint * condition + (1 | Participant_ID)), and the second model included a random slope for condition (DV ~ timepoint * condition + (1 + condition | Participant_ID)). Results from both models revealed that the direction of the effects across conditions and time was consistent with those observed in the repeated ANOVAs.

**Confirmatory analyses: Effects of aesthetic experience on pain.** We found a significant main effect of condition for pain tolerance, pain intensity, and pain affect. The series of ANOVAs were followed up by pairwise comparisons with Bonferroni corrections. For pain tolerance (the maximum time people were allowed to keep their hands in the cold water was 180 s), people tolerated the pain longer in the multimodal and music conditions, compared to the visual art and control conditions. At the same time, multimodal art and music did not differ in their pain tolerating capacity, so the added visual art did not improve the music in this regard. The visual art and control did not differ in their pain tolerating effect, and music alone enabled people tolerate pain longer than visual art (See Table 1 for descriptive data, Table 2 for ANOVA details and Fig 4A for the differences across conditions).

For pain intensity and pain affect (unpleasantness), there was a significant effect of condition, but the effect diminished in the pairwise post hoc comparisons. Thus, although the omnibus tests indicate that at least one condition may differ, the corrected post hoc tests did not identify specific group-level differences (Tables 1, 2, Fig 4B, 4C).

For momentary pain measures at five time points—as mentioned above—we found a time effect (Fig 3A, S4 Table). However, no differences were found depending on the conditions, meaning that the art conditions did not change momentary pain perception during the CPT, nor did they speed up recovery time from the pain.

**Confirmatory analyses: Effects of aesthetic experience on stress.** *Subjective Stress Level:* We assessed subjective momentary stress (by means of VAS) and found no effect of art condition on stress (Fig 3B, S5 Table). While this indicates that the CPT elicits a similar stress response in all four conditions, it also indicates that stress recovery in the different art forms does not differ from the control condition.

*Physiological Measures of Stress (Electrocardiogram & Electrodermal Activity):* The results regarding RMSSD are depicted in Fig 5A, S6 Table, and the results regarding HR are depicted in Fig 5B, S7 Table. We found no effect of condition on RMSSD ($F(1.62, 12.92) = 1.11$, $p = .347$, partial $\eta^2 = 0.12$) nor on HR ($F(2.48, 44.59) = 0.06$, $p = .968$, partial $\eta^2 < 0.001$). The results regarding SCL are depicted in Fig 5C, S8 Table, and we found no effect of condition ($F(2.58, 95.34) = 1.18$, $p = .319$, partial $\eta^2 = .03$). Similarly, we found no interaction effect between condition and time in any of these measures. In an exploratory manner, we have explored EDA in terms of MTVSymp and TVSymp variables [132,133] that are better to detect pain episodes (see more details in Supplementary Material Section 1 for our analysis based on [134,135], and S3 Fig). We found no effect of condition, nor a time and condition interaction.

*Endocrine Measures of Stress (Salivary Stress Biomarkers)*: **Salivary Alpha-Amylase (sAA).** We found that salivary alpha-amylase (sAA) did not differ between conditions ($F(3, 111) = .82$, $p = .49$, partial $\eta^2 = .02$). Similarly, the interaction effect between time and condition was not significant ($F(10.19, 377.17) = 1.04$, $p = .41$, partial $\eta^2 = .03$). This indicates that the CPT elicits similar sAA responses in all four conditions. In addition, this non-significant time x condition interaction effect indicates that the conditions did not differ in terms of recovery, none of the art exposure made people recover from the pain and stress quicker (Fig 6A).

**Salivary Cortisol (sCort).** We found that salivary cortisol (sCort) did not differ between conditions ($F(3, 114) = .70$, $p = .56$, partial $\eta^2 = .02$). Similarly, the interaction effect between time and condition was not significant ($F(6.92, 262.95) =$

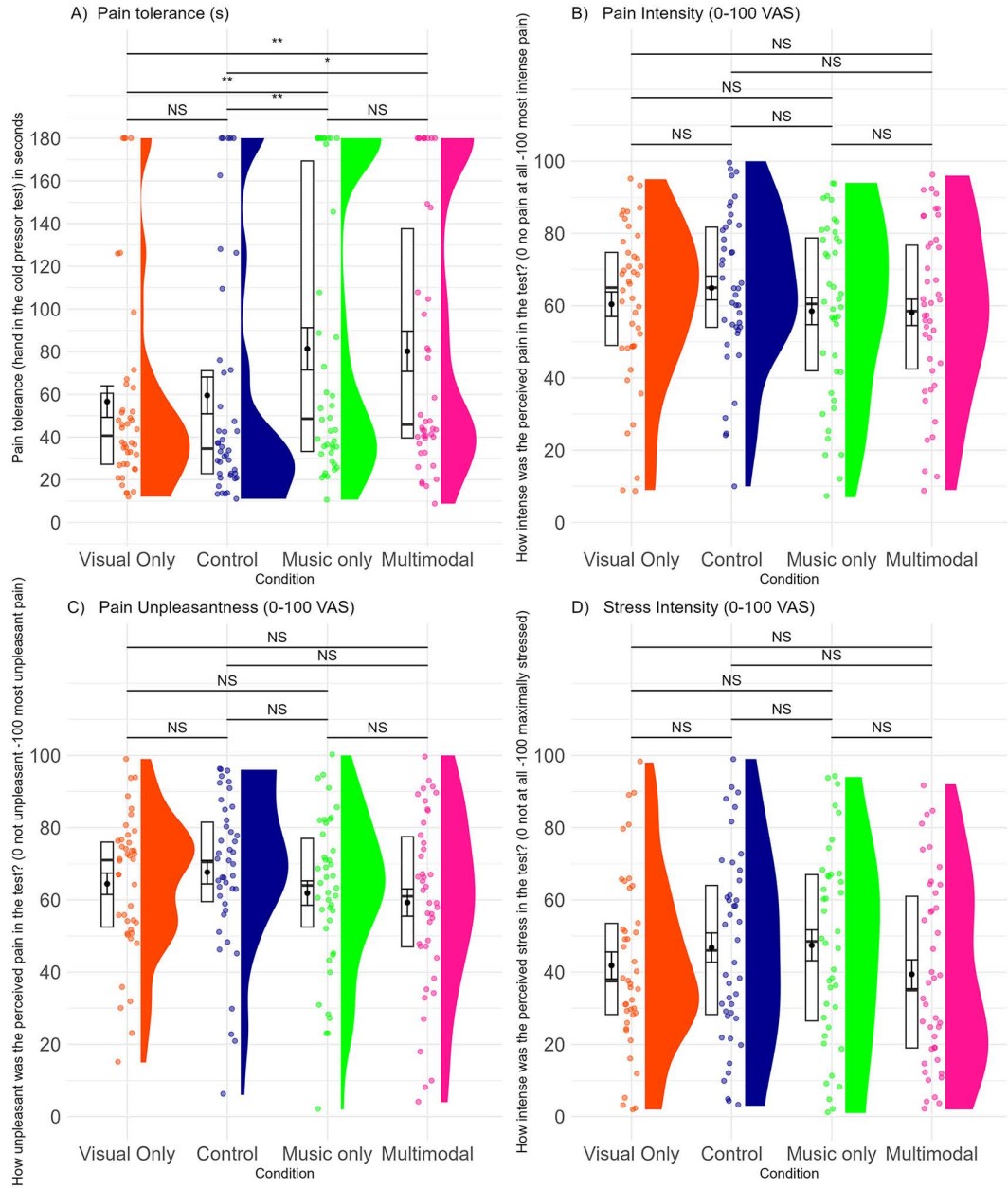

**Fig 4. Pain measures per conditions.** A) Pain tolerance, B) Pain intensity, C) Pain unpleasantness, D) Stress intensity per conditions. Note. A) Pain tolerance was higher with music and multimodal than with visual art or control. Music and multimodal did not differ; visual art and control did not differ; music exceeded visual art. B) Pain intensity, C) Pain Unpleasantness, and D) Stress Intensity did not differ across conditions. The boxplots show the interquartile range without the outliers, the single horizontal line in the boxplots represent the median. The dot in the boxplot represents the mean, and the error bars represent the standard error. The coloured dots show the individual data points, and the half violin plot shows the distribution. Significance levels: NS: Not Significant, $p > 0.05$; *: $p \leq 0.05$; **: $p \leq 0.01$; ***: $p \leq 0.001$; ****: $p \leq 0.0001$. N = 42.

1.53, $p = .16$, partial $\eta^2 = .04$). Similarly to sAA, this non-significant interaction effect indicates that the conditions did not differ in their sCort response to the CPT nor in terms of recovery. None of the art exposure conditions made people recover from the pain and stress quicker (Fig 6B).

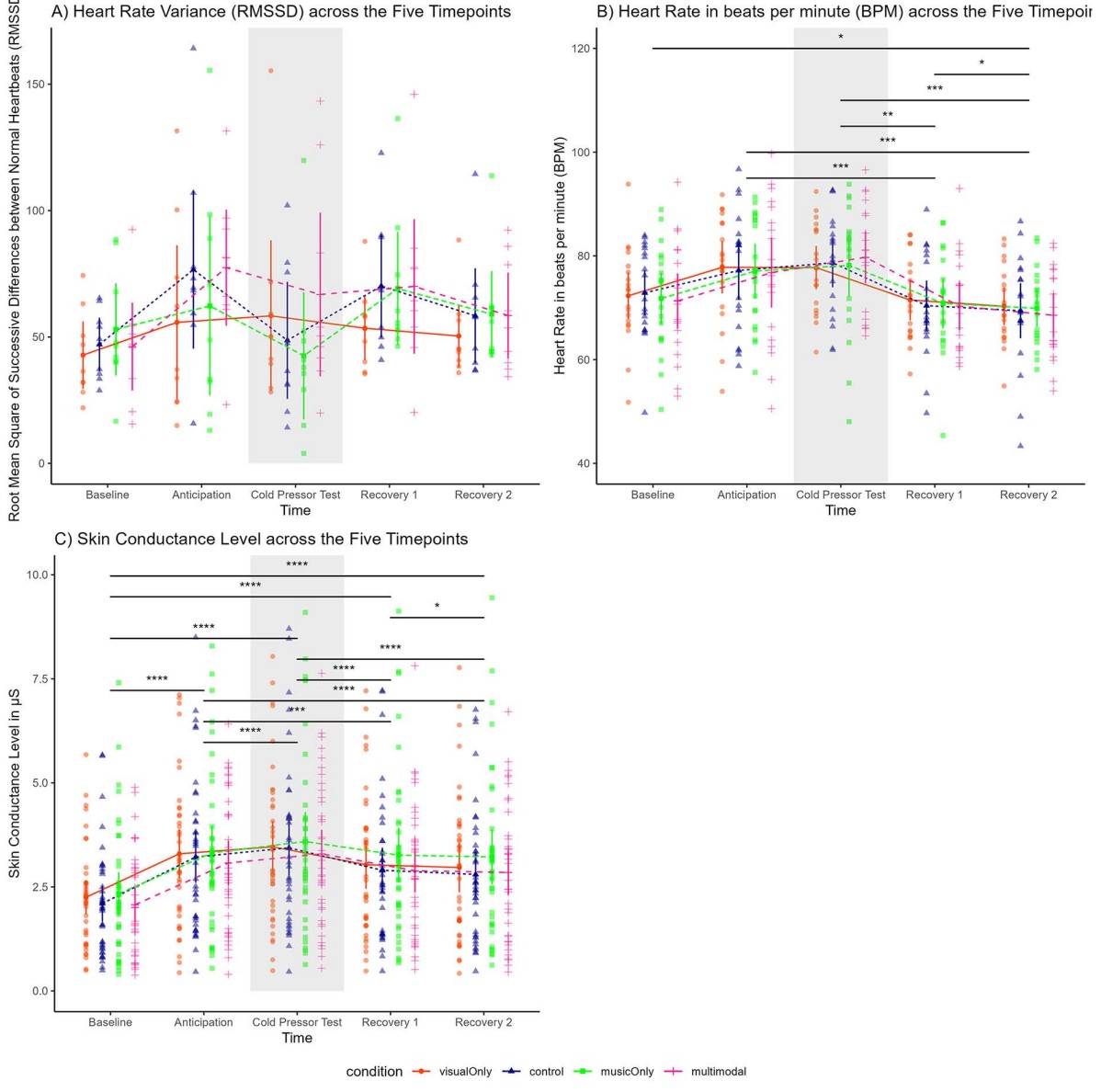

**Fig 5. Physiological measures of Stress.** Note. A) Electrocardiogram in square root of the mean squared differences of successive heartbeat intervals (RMSSD): we did not find an effect of time or condition. B) Heart Rate in beats per minute (BPM): we found a main effect of time, meaning that people experienced reduced heart rate from the CPT to the Recovery 1 and from the CPT to Recovery 2. C) Skin Conductance Level (SCL): we found a main effect of time, indicating that SCL increased from the baseline to the anticipation phase and increased more from the anticipation to the CPT phase due to the stressor, followed by a gradual decrease at the recovery time points. Significance levels: *: $p \leq 0.05$; **: $p \leq 0.01$; ***: $p \leq 0.001$; ****: $p \leq 0.0001$. N = 42.

**Exploratory analyses: Mechanisms of aesthetic experience on pain and stress.** *Aspects of the Experience:* To assess aspects of the experience, we looked at both a lower-level mechanism, i.e., distraction (Fig 7A, S11 Table) and a higher-level mechanism, i.e., mind wandering (Fig 7B, S11 Table). Our findings indicate that all art exposure conditions distracted people's attention from pain ($F(3, 123) = 127.33$, $p < .001$, $\eta^2 = .65$, $BF_{10} = 2.242 \times 10^{38} \pm 0.67\%$, extreme evidence for $H_1$), and made their mind wander ($F(3, 123) = 88.65$, $p < .001$, $\eta^2 = .53$, $BF_{10} = 9.904 \times 10^{28} \pm 0.92\%$, extreme

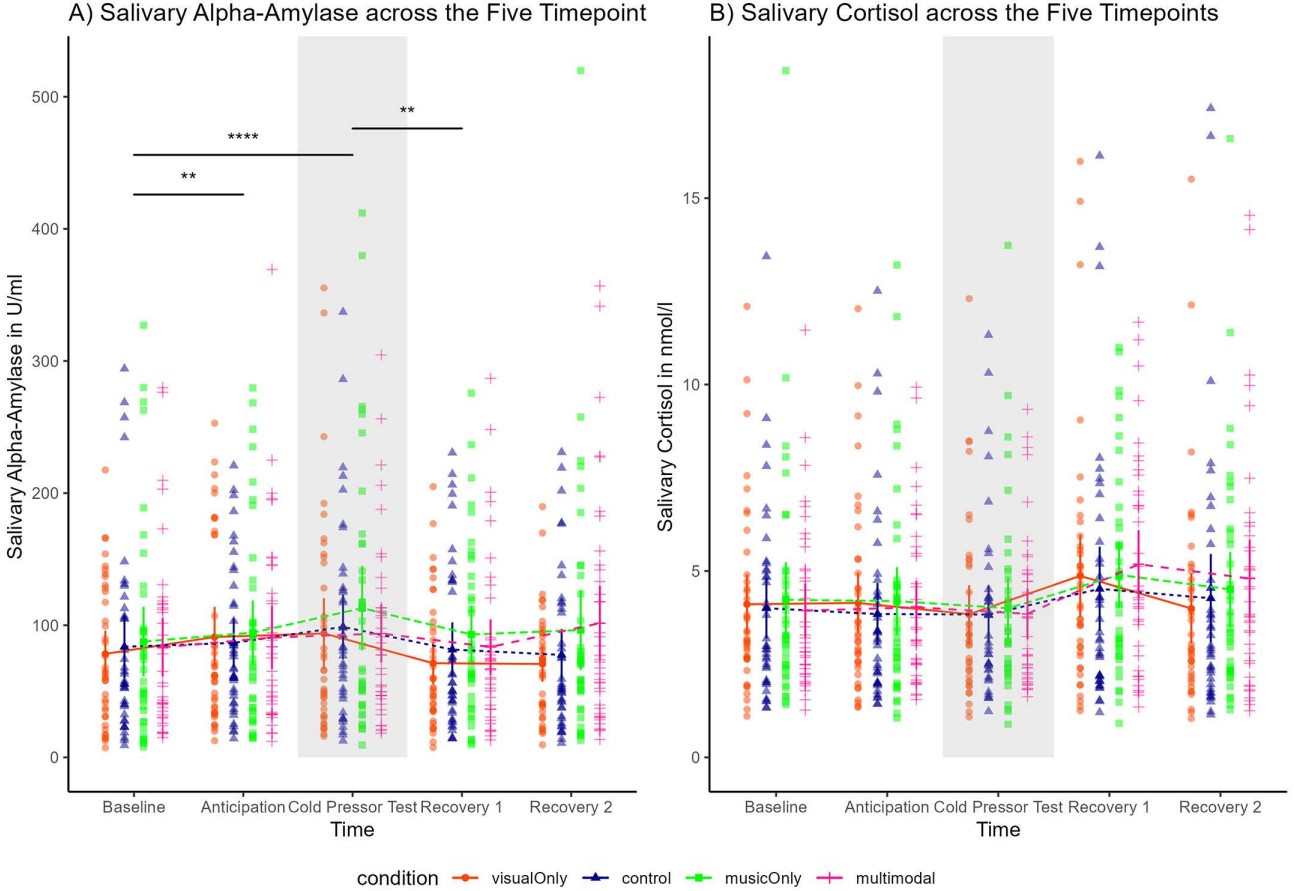

**Fig 6. Salivary alpha-amylase (A) and cortisol (B) across the five timepoints per conditions.** Note. A) For salivary alpha-amylase (sAA), we found effect of time, sAA increased from baseline to anticipation, from baseline to the time point directly after the CPT, and from after the CPT to Recovery 1. B) For salivary cortisol (sCort), we found no effect of time or condition. Significance levels:*: $p \leq 0.05$; **: $p \leq 0.01$; ***: $p \leq 0.001$; ****: $p \leq 0.0001$. N = 42.

evidence for $H_1$), away from the pain, more than a control condition. Furthermore, similar to the effects of condition on pain tolerance, we found that the music and multimodal art conditions did not differ from each other in distraction and mind wandering. However, they both led to more distraction and more mind wandering than the visual only condition. This supports the idea that both distraction and mind wandering are underlying mechanisms for the effect of aesthetic experience on pain. Furthermore, it suggests that the visual only condition was potentially not "strong enough", an issue we return to in the discussion.

Further art features were also different across the conditions, further providing validity that the conditions differed in many aspects, such as beauty, pleasure, enjoyment, liking, nostalgia, personal meaning, chills etc. (S11 and S12 Tables and S2 Fig). Moreover, the general tendency was that all art scored higher in these features than the control condition, and that the pattern observed in the music and multimodal was similar. We note that these mechanisms were exploratory analyses and currently the effect sizes are large as well as the $BF_{10}$ evidence are extreme (S12 Table), and this initial evidence and large effect will potentially decline in future confirmatory studies (see more on the decline effect: [136,137]). Despite these differences between the conditions on art features and mechanisms, the effect did not directly translate to the pain and stress reducing abilities of art.

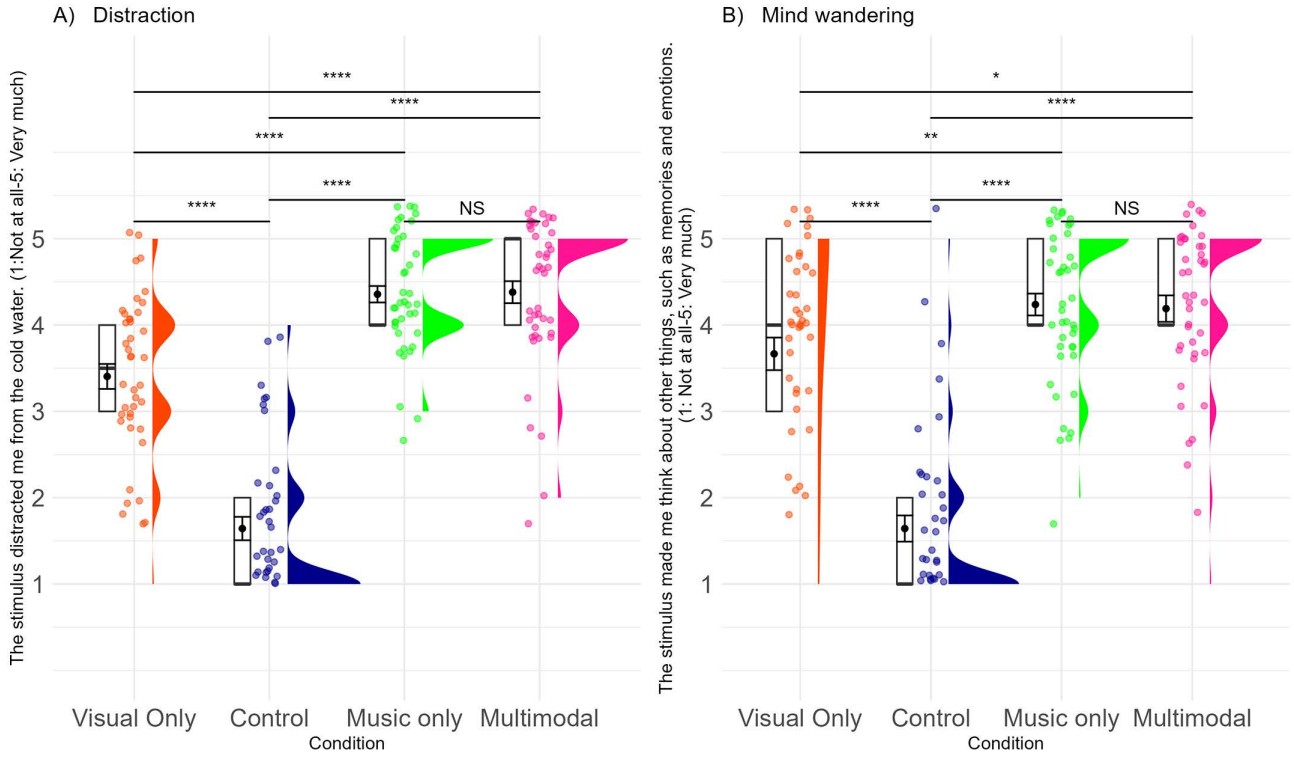

**Fig 7. Mechanisms of Distraction and Mind wandering during the Cold Pressor test per Conditions.** Note. A) Distraction per conditions. All art conditions distracted people's attention from pain. B) Mind wandering per conditions. All art conditions made people's mind wander away from the pain, more than a control condition. Music and multimodal art conditions did not differ from each other in distraction and mind wandering, however, they both led to more distraction and more mind wandering than the visual only condition. The boxplots show the interquartile range without the outliers, the single horizontal line in the boxplots represent the median. The dot in the boxplot represents the mean, and the error bars represent the standard error. The coloured dots show the individual data points, and the half violin plot shows the distribution. Significance levels: NS: Not Significant, $p > 0.05$; *: $p \leq 0.05$; **: $p \leq 0.01$; ***: $p \leq 0.001$; ****: $p \leq 0.0001$. N = 42.

**Motivation of Participants' Art Choices—Qualitative data:** Once participants sent their visual art and music pieces, they were asked to write down why they had chosen the given pairing. We coded this qualitative data to further explore the mechanisms and reveal additional mechanisms that remained underexplored.

After reading the qualitative data, the first author (AF) came up with an initial coding scheme based on the mechanism scales. This initial coding scheme contained the following categories: *Autobiographical memory* (i.e., art reminds people to a memory of their own life), *Feature of the artwork/music piece* (i.e., art was chosen because of its features such as colour, melody, content), *Emotion regulation* (i.e., art makes people feel some emotions such as mood lifting—alternatively, if the participants mentioned that the artwork expressed specific emotions without eliciting emotions in themselves, the latter was named "just emotion" in the coding scheme). Afterwards, two other authors (ES and RM) coded the data based on these categories independently from each other and suggested the inclusion of two new categories: *Meaning Making* (i.e., art made people think about life-related thoughts, meaning of life) and, *Mind Wandering* (i.e., art made people's mind wander to a specific place, triggers imagination). Then, the data was coded again by these two authors (ES and RM). Few discrepancies were found between the coders. The disagreements were resolved though discussion. The coding scheme, the corresponding response options (see Supplementary Material S13 Table), and the coded data can be seen in the OSF page (https://osf.io/yxgrv/) as well as representative quotes and their coding in the Supplementary Material S14 Table.

Based on the responses, we found that most participants chose the art pair because of the features of the artwork/music piece (64.3% of responses contained these elements) as a main motivation. This was followed by autobiographical memories (59.5%) and emotions—either that the art made them feel certain emotions (42.9%), or they associated the art with emotions (31.0%). Note that emotions combined (73.9%) outweighed the art features, or memories. Finally, the last motivation categories were meaning making (38.1%) and mind wandering (16.7%) (Table 3, Fig 8). These results are supporting the results of the mechanisms above, namely distraction and mind wandering.

***Aspects of the Person (Individual Differences)***: To assess aspects of the person, we used the German validated versions of Questionnaire of Cognitive and Affective Empathy (QCAE; [119] and Tellegen Absorption Scale (TAS; [120,121]) to investigate if the traits of empathy and absorption are predictive of benefiting from art exposure on pain and stress. In an exploratory manner, we hypothesized that higher trait cognitive/affective empathy and/or absorption would make people tolerate pain longer in the art conditions (especially multimodal art and music) compared to the control condition.

For empathy, we ran a model including both cognitive and affective empathy for each pain and stress dependent variable separately (i.e., lmer([Dependent Variable] ~ Cognitive Empathy * condition + Affective Empathy * condition + (1|ID)). We only found a main effect ($b = -1.08$, $SE = 0.52$, $t(71.90) = -2.08$, $p = .041$) of cognitive empathy on pain unpleasantness, indicating that people with higher cognitive empathy reported lower pain unpleasantness, independent of condition. No effect of empathy was found on pain tolerance, pain intensity or stress intensity.

We ran similar models for trait absorption (i.e., lmer([Dependent Variable] ~ Trait Absorption * condition + (1 | ID)). We found an interaction effect between trait absorption and the multimodal condition for pain tolerance ($b = 0.58$, $SE = 0.29$, $t(120) = 2.02$, $p = .046$). Specifically, higher trait absorption was associated with increased pain tolerance in the multimodal condition (Fig 9A). Similarly, we found an interaction effect between trait absorption and music only condition for pain intensity ($b = -0.27$, $SE = 0.12$, $t(120) = -2.20$, $p = .030$), showing that higher trait absorption was associated with lower pain intensity in the music-only condition (Fig 9B). No effect of trait absorption was found on pain unpleasantness and stress intensity.

***Combining Aspects of the Experience and Aspects of the Person***: These findings pose challenges for clear interpretation, as the effect of absorption was found on different conditions (once in music, once in multimodal) and on two dependent variables (pain tolerance and pain intensity). Therefore, in an exploratory manner, we investigated whether people higher in absorption were also more distracted and showed more mind wandering, as we assumed that this might be explanation as to why this trait is predictive. Specifically, we tested whether trait absorption is correlated/predicts (in a regression) distraction/mind wandering. We did not find a significant interaction between trait absorption and distraction, meaning that trait absorption does not increase distraction in any of the art conditions. Interestingly, we found a significant interaction between trait absorption and visual only condition ($b = 0.02$, $SE = 0.01$, $t(120) = 2.36$, $p = .020$), meaning that absorption increases mind wandering in the visual only condition (Fig 10). This is especially interesting as people with

**Table 3. Motivation of Participants' Art Choices—Qualitative data Results.**

| Category | Yes | No | NA | Just Emotion |
|---|---|---|---|---|
| Autobiographical memory | 25 (59.5%) | 13 (31.0%) | 4 (9.5%) | – |
| Feature of the artwork/music piece | 27 (64.3%) | 11 (26.2%) | 4 (9.5%) | – |
| Emotion regulation | 18 (42.9%) | 7 (16.6%) | 4 (9.5%) | 13 (31.0%) |
| Meaning making | 16 (38.1%) | 22 (52.4%) | 4 (9.5%) | – |
| Mind wandering | 7 (16.7%) | 31 (73.8%) | 4 (9.5%) | – |

Note. Yes: Person explicitly mentions such elements. No: Person does not mention such element. NA: Cannot be assessed based on the quote. Just Emotion: Person mentions emotions, but it did not necessarily make them feel something. For example, they associate the artwork with these emotions/feel the artwork expresses these emotions.

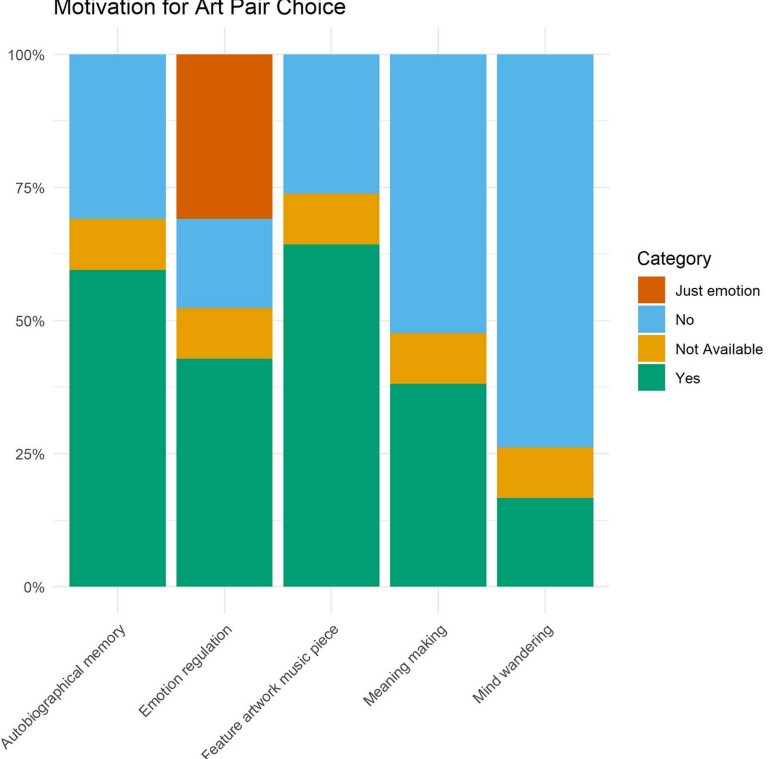

**Fig 8. Motivation for Art Choice.** Note. Most participants have chosen the art pair because of the features of the artwork/music piece (64.3% of responses contained these elements) as a main motivation. This was followed by autobiographical memories (59.5%) and emotions- either that the art made them feel certain emotions (42.9%) or they associated the art with emotions (31.0%). Note that emotions combined (73.9%) overweight the art features or memories. The last motivation categories were meaning making (38.1%) and mind wandering (16.7%). N = 42.

high trait absorption come on the same level of mind wandering in the visual condition as the other conditions, whereas for people with low trait absorption the multimodal or music induced much higher mind wandering.

## Discussion

As pain and stress count among the most important global health challenges, current research is targeting the development and improvement of multimodal pain and stress management programs. While the pain- and stress-reducing effects of music are well researched (e.g., meta-analyses on music and stress [34,128] and music and pain [21]), the effects of visual art, and the combination of both modalities (music and visual art) are much less explored. In a repeated-measures design (music, visual art, multimodal aesthetic experience, control [grey screen]), we tested the (1) pain- and (2) stress-reducing effects of multimodal (music + visual art) aesthetic experience and, in an exploratory manner, (3) investigated underlying mechanisms of aesthetic experience, and (4) individual differences.

First, we expected that (1.1) multimodal (music + visual art) aesthetic experience would reduce pain more than single modal experience and that (1.2) music would reduce pain more than visual art. Our results partly supported these confirmatory hypotheses, both multimodal art and music enabled people tolerate pain longer than single modality of visual art. However, multimodal art and music did not differ. Therefore, our findings suggest that adding music to visual art can make the effect of visual art stronger in the pain influencing capacity, but music is as efficient on its own as multimodal art. For the pain intensity and pain affect measures, we did not find evidence for such pain reducing effect.

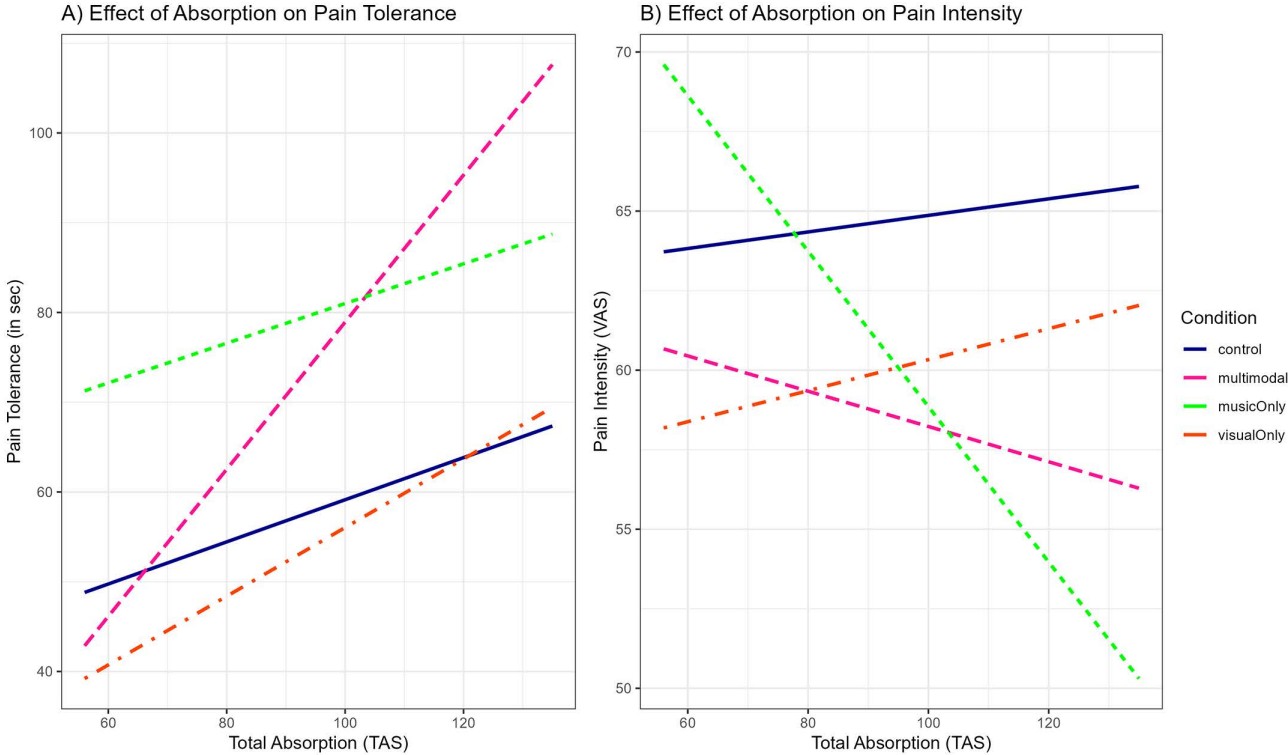

**Fig 9. Effect of Trait Absorption on Pain Tolerance (A) and Pain Intensity (B).** Note. A) Interaction effect between trait absorption and the multimodal condition for Pain Tolerance, specifically, higher trait absorption was associated with increased pain tolerance in the multimodal condition. B) Interaction effect between trait absorption and music only condition for pain intensity, namely higher trait absorption is associated with a decrease in pain intensity in the music-only condition. N = 42.

Second, we expected that (2.1) multimodal (music + visual art) aesthetic experience would reduce activity in stress-responsive systems more than single modal experience; and that (2.2) music would reduce activity in stress-responsive systems more than visual art. We did not find evidence for this confirmatory hypothesis as the art modalities did not differ in their stress influencing capacities.

Third, regarding the exploratory analyses of underlying mechanisms of the experience, we found that any type of art exposure distracted people from the pain and made their mind wander away from the pain and stress – this effect was stronger in both the multimodal and the music condition compared to the visual art condition. Considering that pain tolerance was higher in the two conditions with music than in the condition with only visual art, these results suggest that both distraction and mind wandering may be key for the effect of art on pain tolerance. Visual art may have the same potential to influence pain, but that implementation of the visual art condition might have been "too weak" in terms of inducing distraction and mind wandering, which we discuss in more detail below.

Fourth, regarding the exploratory analyses of underlying mechanisms of the person, we found no effect of trait empathy on pain or stress across conditions, contrasting with prior work suggesting greater music-induced pain relief among individuals with high trait empathy [80]. However, higher cognitive empathy was associated with lower pain unpleasantness, independent of condition. Whereas affective empathy involves sharing others' feelings, cognitive empathy involves understanding them [138]. One possible explanation for our findings might be that individuals higher in cognitive empathy may have focused on understanding the CPT experience and thereby may have cognitively regulated it rather than feeling it. In line with this, Thompson et al. [138] suggested that cognitive empathy relates to better emotion regulation, whereas

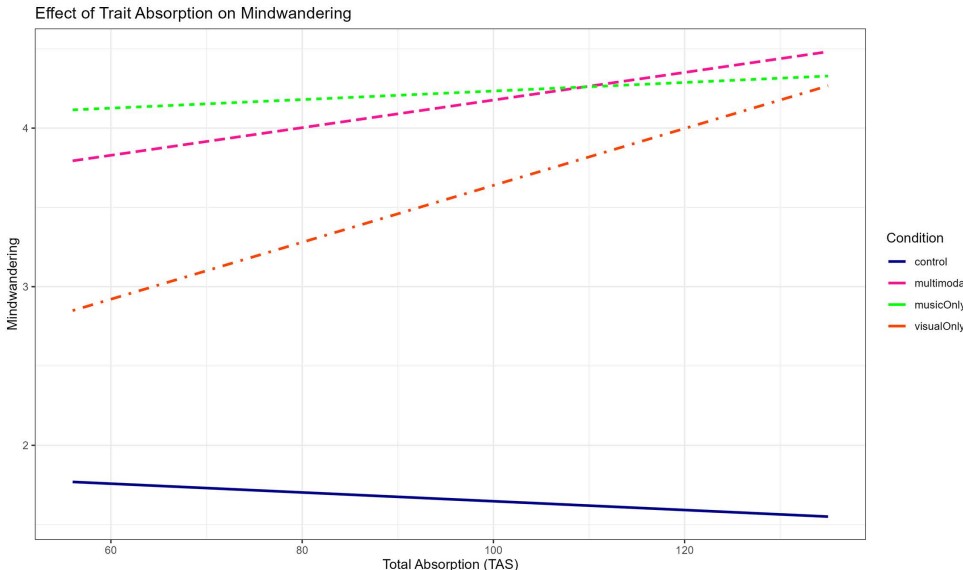

**Fig 10. Effect of Trait Absorption on Mind wandering.** Note. Significant interaction between trait absorption and visual only condition, specifically trait absorption increases mind wandering in the visual only condition. N = 42.

affective empathy relates to greater emotion regulation difficulties. In contrast to trait empathy, we found an interaction effect of trait absorption, for the multimodal condition on pain tolerance and the music condition on pain intensity – both indicate a positive effect of absorption.

To connect these findings of the mechanisms of art experience (distraction, mind wandering) and the role of trait absorption, we tested in an exploratory way whether people higher in absorption would be also more distracted from the pain and have more mind wandering, but we found no evidence for this assumption. However, we did find that trait absorption was associated with mind wandering in the visual only condition. Though only explorative, this analysis suggests that the strength of the manipulation (see also below) might be less relevant for people high in absorption, due to their higher proneness to mind wandering (thus reaching similar levels regardless of condition). Given that mind wandering is predictive of pain tolerance, the findings suggest that people high in absorption may be likely better at tolerating pain (at least when exposed to aesthetic stimuli).

These mechanisms challenge previous studies that found that people with chronic pain do not savour art experience, such as attending opera or theatre [139]. While experiencing chronic pain, people might not have cognitive capacities to self-initiate art engagements. Thus, they might not wish to attend an opera or theatre play, but they might still benefit from low threshold art engagements that distract their mind from the pain, potentially at home, such as listening to music, or looking at (pictures of) visual art. Likewise, this may also pertain to other art categories, such as watching movies or recorded theatre plays. Future studies should continue this line of investigation into the mechanisms. Specifically, Structural Equation Modelling (SEM) would reveal, first, evidence for these separate paths to gain more understanding how trait absorption affects distraction/mind wandering, second, how distraction/ mind wandering, affects pain, and third, how trait absorption affects pain. Due to the complexity of SEM, larger sample sizes are required than what was available in this current study. Therefore, we recommend for future studies to further investigate the underlying mechanisms of art-based pain and stress management with such techniques. It would be especially meaningful to assess other trait measures such as sensitivity for art, aesthetics [140] or music [141], to give more nuanced understanding of individual differences. Importantly, Trupp et al [142] found that those who are more sensitive to art, benefit more from art interventions in terms

of well-being. Similarly, art attendance mediated the relationship between neuroticism and well-being [143]. Beside these mechanisms of the person, future mechanisms of the experience should be disentangled. As we have gained insight into further mechanisms through the qualitative data about people's choices of art pairing, we urge future research to investigate meaning making in a confirmatory analysis. People have reported aspects of thinking about larger life questions, such as the meaning of life, transcendent emotions, reflections, which is in line with the mechanisms found in a recent review investigating through which mechanisms (visual) art can influence well-being [144].

Taken together, these findings suggest that the experience itself seems to be important, that is, art seems to be an effective way to increase pain tolerance, potentially because it may distract from the pain and make people's mind wander away from the pain. This may also explain why the visual art condition was not able to increase pain tolerance as it did increase distraction and mind wandering, but to a lesser extent than music and multimodality. It seems that the music might create a more attention-engaging and emotionally loaded experience which may improve the effect of the visual art in pain and stress experiences. These considerations are based on our exploratory analyses and are in line with findings on aesthetic experience and self-referential processes activating the default mode network [101,145,146]. As such, aesthetic experience integrates sensory and emotional processes related to one's own self [101]. This activates mechanisms in the default mode network, indicating attention inward and not to the outside world, which suggests that attention moves from the outside to the inside. Due to these mechanisms of attentional engagement and mind wandering, art can be a tool to help people to distract themselves from pain and stress experiences. In clinical settings, using the attention-engaging, emotionally charged music may strengthen pain-reducing effects. Importantly, personality traits and gender should be considered. Although emotionally significant, self-selected favourite music was linked to the highest pain tolerance, women who strongly focus on music's emotional aspects ("music empathizers") were highly sensitive to pain when listening to their favourites [147]. Therefore, follow-up studies investigating these mechanisms of the experience and the person are urgently needed before clinical recommendations can be made. One crucial difference may be that music inherently has a time dimension, which might explain why it was more efficient in influencing pain compared to visual art (or any stationary image, such as the control condition of grey screen). For example, it has been shown that the emotional/beauty effect that an art stimulus (visual or auditory) induces, is strongly moderated by the duration of the stimuli [148]. As such, music may exhibit stronger effects related to aspects of the experience, and in turn, stronger effects on pain and stress through the natural duration of the stimulus. This line of argumentation would fit with the previous findings that only music (but not visual art) makes people tolerate (cold) pain longer [36], even though beautiful visual art made people tolerate pain longer than non-beautiful art on CPT [149] and on acute laser evoked pain [35]. Our findings align with two broader frameworks in music research and beyond. First, temporal entrainment, synchronization among independent rhythmic processes [150], may underlie effects unique to music, which has inherent rhythms that can couple with our bodily rhythms, unlike stationary visual art. Second, predictive coding suggests that the brain generates models of the world refined by sensory input, a form of informed "guessing" [151]. Because music is structured by rules that invite predictions and let the listeners anticipate how it will unfold, whereas visual art offers a more static, holistic impression, predictive coding may play a larger role in music than in visual art.

In suggesting that aesthetic quality of the stimulus matters, future studies might investigate this aspect in more detail. Additionally, future research may investigate pain- and stress-reducing effects of a different genre of art, such as films (movie clips have been used in CPT for mood induction, see [152]) as a comparison to music. That said, the multimodal art experience in our study did not add more potential to pain and stress management. This finding replicates our multimodal art study in the museum [153], where no differences were found regarding well-being benefits between the multimodal (music + painting) and the single modality (painting only) conditions.

A related issue is that of stimulus quality. In the visual art condition, we used the digital reproductions of the artworks, and not genuine artworks. Similarly, for the music condition, we used a digital recording—not a live concert. Nevertheless, there might be differences between the quality of a professionally recorded music piece and a photo of an artwork on the

screen. An important difference here is that recorded music pieces (that we used in the study) can be seen as the 'original' work, which cannot be said for the reproductions of artworks we used (see also [154,155]). Therefore, future studies might consider the effect of genuine artworks (i.e., original paintings in the laboratory) on pain perception, which might provide similar extent of distraction or mind wandering like music listening.

Further studies should also consider other control conditions. We used a grey screen with noise cancelling headphones in this study as a control for visual and auditory input. However, our manipulation was self-selected movingly beautiful art that might have personal significance, which causes more automatic processing often without even conscious attention [156]. Future studies should control for personal significance by using a yoked control (e.g., as in [156]) design in which other participants' self-selected stimuli serve as a control stimulus that is unfamiliar to the participant.

In addition, regarding the null effects for our other pain measures, it is possible that attending the pain exposure on different days might have influenced our results. Performing the CPT in a within-subjects-design on different days corresponds to a state-of-the-art procedure [88] with a presumably sufficient length of washout period. However, pain perception might be influenced by many external factors, and conducting pain tests on different days might have introduced a confounding factor. In our other study [149], we repeatedly induced pain on the same day and found that beautiful visual art made people tolerate pain more than non-beautiful art. As in this study, we conducted the sessions on different days due to diurnal changes of cortisol that influence stress within the time of the day; however, from the perspective of pain measurement, the conditions can be recorded in the same day. Future studies should consider implementing a design that can be conducted on the same day, ideally with different types of pain.

Furthermore, especially relevant to our null findings regarding stress, the way stress and pain was induced in our design could potentially be further improved. Specifically—besides pain responses—CPT is commonly used to induce also stress responses (e.g., [114], and indeed our study showed that the CPT was associated with an increase in sAA, which was maintained thereafter. However, differences in sCort were not detected between the conditions. This indicates that while the CPT appears to partly affect autonomic nervous system activity, it may not affect endocrine activity. Similarly, some previous studies have found only limited effects of the CPT on salivary stress biomarkers [157], and a study investigating intra-individual psychological and physiological responses to acute laboratory stressors of different intensities has shown that other laboratory tests are associated with stronger responses in these biomarkers [105]. It is possible that differences may be found between the art conditions in the case of a stronger stressor, which might be investigated further in future studies. Previously, a study on the effect of music on stress response found that while the Trier Social Stress Test (TSST) elicits reliable psychophysiological stress responses, it was too strong for music to elicit psychological stress reduction [32]. The authors discuss that most previous studies have found that music listening does not reduce psychological stress or anxiety when the stressor is too strong but shows stress-reducing effects with milder stressors. In consideration of our results, it seems possible that there might be a dose-relationship here, with the stressor having to be of moderate intensity for music—or art in general—to reduce stress and potentially pain. Furthermore, we note the limitation that our sample size was calculated based on our main behavioural outcome (pain intensity), so physiological and endocrine measures were likely underpowered. Future studies should account for these measures when determining sample size.

Similarly, the design can be improved in terms of physiology data sampling. In our study, we used different time intervals (5 minutes, 20 seconds, 20 seconds, 3 minutes, 3 minutes, see above in the Pain and Stress Manipulation Check section). Future studies should standardize the time intervals for physiological data sampling (i.e., same sampling duration of 2 minutes), as this can help minimize variability. Although we did not find effects of art conditions on stress recovery, neither in terms of subjective stress nor in terms of physiological data—which is in line with the null results in the field of music [128,158,159]—we encourage future studies to improve the art experience (see discussion above about the stimulus quality), design and measurement to test the effect of art on stress recovery.

In addition, our study may be limited in its generalizability, due to having a female-only sample consisting of only Western, Educated, Industrialized, Rich and Democratic (WEIRD; [160]) participants, and the extensive exclusion criteria we

applied (see above) necessary for the accurate measurement of salivary cortisol and alpha-amylase. In general, the more homogenous the sample, the more unclear the extent to which it will generalize. In addition, specifically, it may be that using only females overestimates the found effect, given that females have been found to have higher sensitivity to music [91]. Furthermore, inherently from the nature of laboratory pain studies, there is a sampling bias as to who will volunteer to participate in pain research. Specifically, it is more likely that those people will attend pain studies who have lower level of fear of pain and higher level of sensation-seeking [161]. This could explain why the CPT only partially influenced the saliva stress biomarkers, emphasizing that the effects here should only be interpreted with caution.

Finally, the current state of the literature is limited by a focus on acute pain. As such, it remains an open question if art experiences would have different or similar results for chronic (or other long-term) pain. Similarly, it is theoretically possible that online art viewing (i.e., at the same time with the pain exposure) does not influence pain, but rather exposure to visual art (e.g., in a frame of a museum visit) is able to induce such an effect.

## Supporting information

**S1 Fig. Theory and its Measurements Levels.**
(TIF)

**S2 Fig. Differences between the Conditions in terms of Art Features.**
(TIF)

**S3 Fig. Electrodermal activity in MTVSymp and TVSymp.**
(JPG)

**S1 Table. Summary of research questions, hypotheses, analysis plan and interpretation.**
(DOCX)

**S2 Table. Assessments in the whole course of the study.**
(DOCX)

**S3 Table. Assessments during one testing day (one session).**
(DOCX)

**S4 Table. Momentary Pain VAS according to the Five Time Points.**
(DOCX)

**S5 Table. Momentary Stress VAS according to the Five Time Points.**
(DOCX)

**S6 Table. Heart Rate Variability in Square Root of the Mean Squared Differences of Successive Heartbeat Intervals (RMSSD) according to the Five Time Points.**
(DOCX)

**S7 Table. Heart Rate in Beat Per Minute (BPM) according to the Five Time Points.**
(DOCX)

**S8 Table. Skin Conductance Level (SCL) in μS according to the Five Time Points.**
(DOCX)

**S9 Table. Salivary alpha-amylase (sAA) [U/ml] according to the Five Time Points.**
(DOCX)

**S10 Table. Salivary cortisol (sCort) [nmol/l] according to the Five Time Points.**
(DOCX)

**S11 Table. Descriptive Data of the Mechanisms of Aesthetic Experience on Pain and Stress as well as Features of the Art.**
(DOCX)

**S12 Table. ANOVAs of Mechanisms of Pain and Stress as well as Features of the Arts.**
(DOCX)

**S13 Table. Qualitative Data Coding Scheme for Pairs of Visual Artworks and Music Pieces.**
(DOCX)

**S14 Table. Representative Quotes and their Coding.**
(DOCX)

## Acknowledgments

The authors would like to thank Prof. Ki H. Chon and Dr. Youngsun Kong for providing the code for calculating TVSymp and Dr. Alexander Nicholls for English proof reading of the manuscript. Furthermore, we thank Sasa Vladuljevic, Kira Körber, Laura Roppelt and Maxie Labonte for the data collection and for Markus Foramitti for proofreading earlier versions of this manuscript.

Change in authorship compared to the Registered Report Protocol: Andreas Gartus was added to the author list.

## Author contributions

**Conceptualization:** Anna Fekete, Rosa M. Maidhof, Eva Specker, Helmut Leder.

**Data curation:** Anna Fekete, Rosa M. Maidhof.

**Formal analysis:** Anna Fekete, Andreas Gartus.

**Funding acquisition:** Anna Fekete, Rosa M. Maidhof.

**Investigation:** Anna Fekete, Rosa M. Maidhof, Eva Specker.

**Methodology:** Anna Fekete, Rosa M. Maidhof, Eva Specker, Andreas Gartus.

**Project administration:** Anna Fekete, Rosa M. Maidhof.

**Resources:** Urs M. Nater.

**Supervision:** Urs M. Nater, Helmut Leder.

**Visualization:** Anna Fekete.

**Writing – original draft:** Anna Fekete, Rosa M. Maidhof, Eva Specker, Urs M. Nater, Helmut Leder.

**Writing – review & editing:** Anna Fekete, Rosa M. Maidhof, Eva Specker, Andreas Gartus, Urs M. Nater, Helmut Leder.

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
