## [Decision Letter · Decision Letter 0]

13 Jan 2026

PONE-D-25-45648Registered Report: How does Art Impact Pain and Stress? Exposure to Multimodal Art (Music + Visual) and Music Alone Enhances Pain Tolerance More Than Visual Art, but Neither Art Form Impacts Autonomic or Endocrine MarkersPLOS One

Dear Dr. Fekete,

Thank you for submitting your manuscript to PLOS ONE. After careful consideration, we feel that it has merit but does not fully meet PLOS ONE’s publication criteria as it currently stands. Therefore, we invite you to submit a revised version of the manuscript that addresses the points raised during the review process.

Please respond to all comments and highlight the changes in the ms.

If applicable, we recommend that you deposit your laboratory protocols in protocols.io to enhance the reproducibility of your results. Protocols.io assigns your protocol its own identifier (DOI) so that it can be cited independently in the future. For instructions see: https://journals.plos.org/plosone/s/submission-guidelines#loc-laboratory-protocols. Additionally, PLOS ONE offers an option for publishing peer-reviewed Lab Protocol articles, which describe protocols hosted on protocols.io. Read more information on sharing protocols at . Additionally, PLOS ONE offers an option for publishing peer-reviewed Lab Protocol articles, which describe protocols hosted on protocols.io. Read more information on sharing protocols at https://plos.org/protocols?utm_medium=editorial-email&utm_source=authorletters&utm_campaign=protocols..

We look forward to receiving your revised manuscript.

Kind regards,

Thiago P. Fernandes, PhD

Academic Editor

PLOS One

Journal Requirements:

“This research was funded by two grants to Anna Fekete and to Rosa Maidhof, who both received a grant (“Förderstipendium”) from the University of Vienna, Büro Studienpräses.”

“This research was funded by two grants to Anna Fekete and to Rosa Maidhof, who both received a grant (“Förderstipendium”) from the University of Vienna, Büro Studienpräses.”

5. Please remove your figures from within your manuscript file, leaving only the individual TIFF/EPS image files, uploaded separately. These will be automatically included in the reviewers’ PDF.

Reviewers' comments:

Reviewer's Responses to Questions

**Comments to the Author**

1. Does the manuscript adhere to the experimental procedures and analyses described in the Registered Report Protocol?

If the manuscript reports any deviations from the planned experimental procedures and analyses, those must be reasonable and adequately justified.

Reviewer #1: Yes

Reviewer #2: Yes

2. If the manuscript reports exploratory analyses or experimental procedures not outlined in the original Registered Report Protocol, are these reasonable, justified and methodologically sound?

A Registered Report may include valid exploratory analyses not previously outlined in the Registered Report Protocol, as long as they are described as such.

Reviewer #1: Yes

Reviewer #2: Partly

3. Are the conclusions supported by the data and do they address the research question presented in the Registered Report Protocol?

The manuscript must describe a technically sound piece of scientific research with data that supports the conclusions. The conclusions must be drawn appropriately based on the research question(s) outlined in the Registered Report Protocol and on the data presented.

Reviewer #1: Yes

Reviewer #2: Partly

4. Have the authors made all data underlying the findings in their manuscript fully available?

Reviewer #1: Yes

Reviewer #2: Yes

5. Is the manuscript presented in an intelligible fashion and written in standard English?

*PLOS ONE* does not copyedit accepted manuscripts, so the language in submitted articles must be clear, correct, and unambiguous. Any typographical or grammatical errors should be corrected at revision, so please note any specific errors here.does not copyedit accepted manuscripts, so the language in submitted articles must be clear, correct, and unambiguous. Any typographical or grammatical errors should be corrected at revision, so please note any specific errors here.

Reviewer #1: Yes

Reviewer #2: Yes

6. Review Comments to the Author

Please use the space provided to explain your answers to the questions above. (Please upload your review as an attachment if it exceeds 20,000 characters)

Reviewer #1: General Evaluation

The manuscript presents a carefully designed and methodologically rigorous registered report examining how multimodal (music + visual art) aesthetic experiences influence experimentally induced pain and stress. The authors have implemented a well-controlled within-subjects design, preregistered their procedures, and ensured full data transparency via OSF (https://osf.io/yxgrv/).

The study makes a valuable contribution to empirical aesthetics and psychophysiology by demonstrating that multimodal and music-only conditions significantly enhance pain tolerance compared with visual art and control, while physiological stress markers remain unchanged. The paper is detailed, intellectually rich, and methodologically transparent.

1. Assessment of the Literature Review

The literature review is generally comprehensive, well-structured, and logically organized around three major research areas: (1) music and pain/stress, (2) visual art and pain/stress, and (3) multimodal or combined art experiences.

Strengths:

The authors accurately synthesize core meta-analyses and seminal studies (e.g., Lee, 2016; de Witte et al., 2020; Mitchell et al., 2008; de Tommaso et al., 2008).

The review effectively links aesthetic engagement with psychological mechanisms of pain modulation (attention, emotion, meaning-making), referencing Howlin & Rooney (2020).

The authors appropriately identify inconsistencies in prior findings on visual art and position their work as addressing these gaps.

Areas for improvement:

The review could benefit from the inclusion of more recent work in multimodal and neuroaesthetic research, such as Brattico & Vuust (2017) and Koelsch (2020), to reinforce the neural and cognitive basis for cross-modal integration.

The discussion of individual difference variables (trait absorption and empathy) is thoughtful but could be expanded by referencing newer frameworks, such as Aesthetic Responsiveness (Schlotz et al., 2020) or Musical Aesthetic Sensitivity (Clemente & Nadal, 2022), which would contextualize these traits within contemporary personality–aesthetics research.

The introduction may briefly address the growing literature on digital versus in-person art experiences (e.g., Specker et al., 2023), since the present study used digital reproductions of artworks.

The number of citations is extensive and might be streamlined by emphasizing recent meta-analyses to enhance readability.

Overall, the literature review is well-developed and conceptually solid, warranting only minor updates to incorporate the latest findings in multimodal and individual-difference research.

2. Study Rationale and Theoretical Framing

The rationale is strong and socially relevant. The authors clearly outline the public health importance of nonpharmacological pain management and the potential of aesthetic experiences. However, the theoretical link to the Vienna Integrated Model of Art Perception (VIMAP) could be elaborated further to clarify how the hypothesized mechanisms (distraction, mind wandering, absorption) align with the model’s top–down and bottom–up components.

3. Methods and Procedures

The within-subject design and counterbalancing are exemplary. The authors report methodological details with commendable precision, including apparatus, physiological recordings, and saliva sampling.

Clarification is recommended regarding whether color and luminance calibration were standardized for the visual stimuli, as this can influence perceived beauty and emotional impact.

The female-only, non-artist sample is justified for experimental control, yet the authors should explicitly acknowledge its limitation for generalizability.

Providing mean ± SD for immersion durations in each condition would enhance interpretability of the pain-tolerance results.

4. Results and Interpretation

The results are internally consistent and presented transparently. The primary findings—greater pain tolerance under music and multimodal conditions—are statistically robust and clearly reported.

The absence of significant differences in physiological stress indices is plausible and well discussed, but a note regarding statistical power for endocrine measures (given high individual variability) would be helpful.

The exploratory analyses on distraction and mind wandering are valuable and could be strengthened with a simplified conceptual figure illustrating their proposed mediating roles.

The qualitative data analysis of participants’ motivations adds richness to the manuscript; including one or two representative quotes in the Supplementary Information would provide readers with a more tangible sense of the responses.

5. Discussion and Implications

The Discussion is comprehensive and integrates findings across multiple domains. However, the first three paragraphs (pp. 46–48) contain overlapping statements regarding Hypotheses 1–2 and could be condensed for readability.

The authors’ interpretation of trait absorption as a significant moderator is novel and insightful. The null finding for empathy could be further contextualized by distinguishing between cognitive and affective empathy and their differing relevance for aesthetic engagement.

The discussion of music’s temporal dynamics as an explanatory factor is compelling; linking this to theoretical frameworks such as temporal entrainment or predictive coding would enrich the argument.

In conclusion, the authors may wish to expand on the applied implications, highlighting how art-based interventions could be adapted for clinical pain management or mental health contexts.

6. Figures, Tables, and Presentation

Figures 3–7 are clear and well designed. The authors should ensure all legends include n values and that statistical annotations remain consistent throughout.

Minor typographical and stylistic issues (e.g., inconsistent spacing in Table 2, repeated use of “aesthetic experience”) should be corrected during revision.

Overall, the manuscript is clearly written and requires only light language editing.

Summary Recommendation

The manuscript is technically sound, conceptually coherent, and methodologically transparent. The literature review is comprehensive but would benefit from modest updates to reflect recent multimodal and individual-difference research.

Reviewer #2: Major Comments

1. Abstract: lack of quantitative results

The Abstract does not report any quantitative information (e.g., means, effect sizes, confidence intervals) and therefore does not adequately convey the empirical strength of the findings. Given that most outcomes showed no differences across conditions, a more quantitative and transparent abstract is particularly important for PLOS One readers. The authors should revise the Abstract to clearly report key effect sizes and null results.

2. Length and structure of the manuscript

The manuscript is considerably longer than necessary relative to the core empirical contribution. The Introduction and theoretical sections are highly detailed and at times repetitive, particularly in the discussion of mechanisms and multimodal aesthetics. Substantial shortening is recommended, especially by reducing theoretical elaboration that is not directly tested or supported by the reported results. Improving focus would greatly enhance readability and impact.

3. Clarity regarding outcomes and analysis strategy

Although the Methods section is very detailed, it is not always clear which outcomes constitute primary confirmatory endpoints versus secondary or exploratory analyses. Given the Registered Report format, this distinction is crucial. The authors should more explicitly identify primary outcomes and hypotheses in the main text and ensure that the Results and Discussion are structured accordingly.

4. Interpretation of null findings and discussion scope

The study reports largely null effects for autonomic and endocrine stress markers. While these findings are valid and important, the Discussion sometimes extrapolates mechanistic interpretations that are not clearly supported by the physiological data. The authors should adopt a more restrained interpretation and ensure that conclusions closely track the empirical evidence, particularly regarding proposed underlying mechanisms.

5. Generalizability and framing of implications

The study sample is highly specific (young, healthy female participants with strict exclusion criteria). While this is methodologically justified, the implications for broader pain and stress management should be framed more cautiously. Claims regarding potential applications or general benefits of art exposure should be explicitly limited to the studied population and context.

I have not provided minor, line-by-line comments, as the key issues concern structure, focus, and interpretation rather than surface-level edits.

7. PLOS authors have the option to publish the peer review history of their article (what does this mean?). If published, this will include your full peer review and any attached files.). If published, this will include your full peer review and any attached files.

.

Reviewer #1: No

Reviewer #2: No

---

## [Author Response · Author response to Decision Letter 1]

26 Feb 2026

We have uploaded a separate Response to Reviewers file. For your convenience, we have also included the responses below.

Manuscript number: PONE-D-25-45648

Title: Registered Report: How does Art Impact Pain and Stress? Exposure to Multimodal Art (Music + Visual) and Music Alone Enhances Pain Tolerance More Than Visual Art, but Neither Art Form Impacts Autonomic or Endocrine Markers

Dear Editor,

Thank you for the opportunity of resubmitting our manuscript.

Dear Editor, Dear Reviewers,

Below we respond first to the comments of the editor, followed by the responses to the reviewers (written in bold and italics). Note all page numbers indicated in the response letter are based on the manuscript without tracked changes.

Response to the Editor:

Journal Requirements:

Thank you very much. We have ensured that our manuscript meets PLOS ONE's style requirements, including those for file naming.

“This research was funded by two grants to Anna Fekete and to Rosa Maidhof, who both received a grant (“Förderstipendium”) from the University of Vienna, Büro Studienpräses.”

Thank you for your note about the funding. We have addressed this point in the Cover Letter.

The grants that we had received is an internal university financial support grant and were without an official reference number.

“This research was funded by two grants to Anna Fekete and to Rosa Maidhof, who both received a grant (“Förderstipendium”) from the University of Vienna, Büro Studienpräses.”

Thank you for this comment. We addressed it in the Cover Letter.

5. Please remove your figures from within your manuscript file, leaving only the individual TIFF/EPS image files, uploaded separately. These will be automatically included in the reviewers’ PDF.

We removed our figures from within our manuscript files and left the individual image files separately.

Thank you, we reviewed all recommendations to cite specific previously published works and only added them if we found them relevant to our work.

1. Does the manuscript adhere to the experimental procedures and analyses described in the Registered Report Protocol?

If the manuscript reports any deviations from the planned experimental procedures and analyses, those must be reasonable and adequately justified.

Reviewer #1: Yes

Reviewer #2: Yes

2. If the manuscript reports exploratory analyses or experimental procedures not outlined in the original Registered Report Protocol, are these reasonable, justified and methodologically sound?

A Registered Report may include valid exploratory analyses not previously outlined in the Registered Report Protocol, as long as they are described as such.

Reviewer #1: Yes

Reviewer #2: Partly

3. Are the conclusions supported by the data and do they address the research question presented in the Registered Report Protocol?

The manuscript must describe a technically sound piece of scientific research with data that supports the conclusions. The conclusions must be drawn appropriately based on the research question(s) outlined in the Registered Report Protocol and on the data presented.

Reviewer #1: Yes

Reviewer #2: Partly

4. Have the authors made all data underlying the findings in their manuscript fully available?

Reviewer #1: Yes

Reviewer #2: Yes

5. Is the manuscript presented in an intelligible fashion and written in standard English?

Reviewer #1: Yes

Reviewer #2: Yes

Dear Reviewers,

Thank you for taking the time to review our manuscript and for your constructive feedback.

Response to Reviewer 1:

Reviewer #1: General Evaluation

The manuscript presents a carefully designed and methodologically rigorous registered report examining how multimodal (music + visual art) aesthetic experiences influence experimentally induced pain and stress. The authors have implemented a well-controlled within-subjects design, preregistered their procedures, and ensured full data transparency via OSF (https://osf.io/yxgrv/).

The study makes a valuable contribution to empirical aesthetics and psychophysiology by demonstrating that multimodal and music-only conditions significantly enhance pain tolerance compared with visual art and control, while physiological stress markers remain unchanged. The paper is detailed, intellectually rich, and methodologically transparent.

1. Assessment of the Literature Review

The literature review is generally comprehensive, well-structured, and logically organized around three major research areas: (1) music and pain/stress, (2) visual art and pain/stress, and (3) multimodal or combined art experiences.

Strengths:

The authors accurately synthesize core meta-analyses and seminal studies (e.g., Lee, 2016; de Witte et al., 2020; Mitchell et al., 2008; de Tommaso et al., 2008).

The review effectively links aesthetic engagement with psychological mechanisms of pain modulation (attention, emotion, meaning-making), referencing Howlin & Rooney (2020).

The authors appropriately identify inconsistencies in prior findings on visual art and position their work as addressing these gaps.

We thank the reviewer for the positive assessment of our manuscript.

Areas for improvement:

The review could benefit from the inclusion of more recent work in multimodal and neuroaesthetic research, such as Brattico & Vuust (2017) and Koelsch (2020), to reinforce the neural and cognitive basis for cross-modal integration.

The discussion of individual difference variables (trait absorption and empathy) is thoughtful but could be expanded by referencing newer frameworks, such as Aesthetic Responsiveness (Schlotz et al., 2020) or Musical Aesthetic Sensitivity (Clemente & Nadal, 2022), which would contextualize these traits within contemporary personality–aesthetics research.

The introduction may briefly address the growing literature on digital versus in-person art experiences (e.g., Specker et al., 2023), since the present study used digital reproductions of artworks.

The number of citations is extensive and might be streamlined by emphasizing recent meta-analyses to enhance readability.

Overall, the literature review is well-developed and conceptually solid, warranting only minor updates to incorporate the latest findings in multimodal and individual-difference research.

Thank you for the suggestion. As a Registered Report, our Stage 2 Introduction and Methods intentionally mirror the approved Stage 1 protocol (https://doi.org/10.1371/journal.pone.0266545), following guidance (https://doi.org/10.1371/journal.pcbi.1010571) to limit changes to tense shifts and factual corrections. Our academic editor confirmed this approach: keep these sections largely unchanged, with minor clarifications; substantial shortening is discouraged. For PLOS ONE examples, you may see: https://journals.plos.org/plosone/search?filterJournals=PLoSONE&filterArticleTypes=Research%20Article&q=registered%20report%20protocol&page=1 ´.

Therefore, we do not have the opportunity to implement the suggested references.

Regarding the comment on digital reproductions, we discuss this issue in our Discussion section:

“A related issue is that of stimulus quality. In the visual art condition, we used the digital reproductions of the artworks, and not genuine artworks. Similarly, for the music condition, we used a digital recording—not a live concert. Nevertheless, there might be differences between the quality of a professionally recorded music piece and a photo of an artwork on the screen. An important difference here is that recorded music pieces (that we used in the study) can be seen as the ‘original’ work, which cannot be said for the reproductions of artworks we used (see also [150, 151]. Therefore, future studies might consider the effect of genuine artworks (i.e., original paintings in the laboratory) on pain perception, which might provide similar extent of distraction or mind wandering like music listening.” (p.46)

2. Study Rationale and Theoretical Framing

The rationale is strong and socially relevant. The authors clearly outline the public health importance of nonpharmacological pain management and the potential of aesthetic experiences. However, the theoretical link to the Vienna Integrated Model of Art Perception (VIMAP) could be elaborated further to clarify how the hypothesized mechanisms (distraction, mind wandering, absorption) align with the model’s top–down and bottom–up components.

Thank you, indeed this could have been a good suggestion for the Stage 1 review, but unfortunately, we cannot make changes in the literature review due to the PLOS ONE guidelines associated with Registered Reports (see also comment 1).

3. Methods and Procedures

The within-subject design and counterbalancing are exemplary. The authors report methodological details with commendable precision, including apparatus, physiological recordings, and saliva sampling.

Clarification is recommended regarding whether color and luminance calibration were standardized for the visual stimuli, as this can influence perceived beauty and emotional impact.

We did not standardize color or luminance for the stimulus images. Stimuli were presented using PsychoPy and scaled proportionally to fill the display, but no color or luminance adjustments were applied.

For our stimulus set, luminance standardization was not appropriate: enforcing uniform luminance across diverse content (e.g., night scenes vs. snowy landscapes) would produce unrealistic, desaturated images, potentially altering the very aesthetic and emotional qualities under study.

The same monitor was used for all participants, so the calibration was consistent. The monitors were not hardware-calibrated. We note that calibration is more critical for paradigms where precise luminance control affects performance (e.g., tasks requiring fast reaction times), rather than studies prioritizing ecological validity of complex, naturalistic stimuli.

In summary, we do not think that this was a limitation of the study as we prioritized ecological validity in the present study and therefore, we did not modify the stimuli material.

The female-only, non-artist sample is justified for experimental control, yet the authors should explicitly acknowledge its limitation for generalizability.

We agree with the reviewer on this limitation.

We discuss this limitation:

“In addition, our study may be limited in its generalizability, due to having a female-only sample consisting of only Western, Educated, Industrialized, Rich and Democratic (WEIRD; [156]) participants, and the extensive exclusion criteria we applied (see above) necessary for the accurate measurement of salivary cortisol and alpha-amylase. In general, the more homogenous the sample, the more unclear the extent to which it will generalize. In addition, specifically, it may be that using only females overestimates the found effect, given that females have been found to have higher sensitivity to music [91]. Furthermore, inherently from the nature of laboratory pain studies, there is a sampling bias as to who will volunteer to participate in pain research. Specifically, it is more likely that those people will attend pain studies who have lower level of fear of pain and higher level of sensation-seeking [157]. This could explain why the CPT only partially influenced the saliva stress biomarkers, emphasizing that the effects here should only be interpreted with caution.” (p. 48-49)

Providing mean ± SD for immersion durations in each condition would enhance interpretability of the pain-tolerance results.

We would like to note that people immersed their hands in cold water, and this immersion time serves as a measure of pain tolerance (see more details on this measure in the Pain measures section, p.22).

We have included the mean and SD of immersion duration = pain tolerance (in s) to Table 1. Descriptive of Pain and Stress Measures: Pain tolerance (in s): Visual M=56.63 (SD=47.86), Control M= 59.50 (SD=55.86), Music M =81.34 (SD=64.19), Multimodal M= 80.19 (SD=61.05) (p.32-33).

4. Results and Interpretation

The results are internally consistent and presented transparently. The primary findings—greater pain tolerance under music and multimodal conditions—are statistically robust and clearly reported.

We would like to thank the reviewer for the overall positive evaluation of our results and their interpretation.

The absence of significant differences in physiological stress indices is plausible and well discussed, bu

---

## [Decision Letter · Decision Letter 1]

15 Mar 2026

PONE-D-25-45648R1Registered Report: How does Art Impact Pain and Stress?

Exposure to Multimodal Art (Music + Visual) and Music Alone Enhances Pain Tolerance More Than Visual Art, but Neither Art Form Impacts Autonomic or Endocrine MarkersPLOS One

Dear Dr. Fekete,

Thank you for submitting your manuscript to PLOS ONE. After careful consideration, we feel that it has merit but does not fully meet PLOS ONE’s publication criteria as it currently stands. Therefore, we invite you to submit a revised version of the manuscript that addresses the points raised during the review process.

 Please respond the comments and highlight all changes in the revised manuscript.Wishing you success with the study.

If applicable, we recommend that you deposit your laboratory protocols in protocols.io to enhance the reproducibility of your results. Protocols.io assigns your protocol its own identifier (DOI) so that it can be cited independently in the future. For instructions see: https://journals.plos.org/plosone/s/submission-guidelines#loc-laboratory-protocols. Additionally, PLOS ONE offers an option for publishing peer-reviewed Lab Protocol articles, which describe protocols hosted on protocols.io. Read more information on sharing protocols at . Additionally, PLOS ONE offers an option for publishing peer-reviewed Lab Protocol articles, which describe protocols hosted on protocols.io. Read more information on sharing protocols at https://plos.org/protocols?utm_medium=editorial-email&utm_source=authorletters&utm_campaign=protocols..

We look forward to receiving your revised manuscript.

Kind regards,

Thiago P. Fernandes, PhD

Academic Editor

PLOS One

Journal Requirements:

Reviewers' comments:

Reviewer's Responses to Questions

**Comments to the Author**

1. Does the manuscript adhere to the experimental procedures and analyses described in the Registered Report Protocol?

If the manuscript reports any deviations from the planned experimental procedures and analyses, those must be reasonable and adequately justified.

Reviewer #1: Yes

Reviewer #2: Yes

2. If the manuscript reports exploratory analyses or experimental procedures not outlined in the original Registered Report Protocol, are these reasonable, justified and methodologically sound?

A Registered Report may include valid exploratory analyses not previously outlined in the Registered Report Protocol, as long as they are described as such.

Reviewer #1: Yes

Reviewer #2: Partly

3. Are the conclusions supported by the data and do they address the research question presented in the Registered Report Protocol?

The manuscript must describe a technically sound piece of scientific research with data that supports the conclusions. The conclusions must be drawn appropriately based on the research question(s) outlined in the Registered Report Protocol and on the data presented.

Reviewer #1: Yes

Reviewer #2: Partly

4. Have the authors made all data underlying the findings in their manuscript fully available?

Reviewer #1: Yes

Reviewer #2: Yes

5. Is the manuscript presented in an intelligible fashion and written in standard English?

*PLOS ONE* does not copyedit accepted manuscripts, so the language in submitted articles must be clear, correct, and unambiguous. Any typographical or grammatical errors should be corrected at revision, so please note any specific errors here.does not copyedit accepted manuscripts, so the language in submitted articles must be clear, correct, and unambiguous. Any typographical or grammatical errors should be corrected at revision, so please note any specific errors here.

Reviewer #1: Yes

Reviewer #2: Yes

6. Review Comments to the Author

Please use the space provided to explain your answers to the questions above. (Please upload your review as an attachment if it exceeds 20,000 characters)

Reviewer #1: Reviewer Comments to the Authors

General Evaluation

I would like to thank the authors for their thorough and thoughtful response to my previous comments. The revised manuscript has been significantly strengthened, and the authors have demonstrated a commendable level of scientific rigor in addressing each of the concerns raised. This Stage 2 Registered Report now provides a highly transparent and well-structured presentation of the findings regarding the impact of art on pain and stress. I am particularly satisfied with how the authors managed the balance between the inherent constraints of the Registered Report format and the need for greater empirical clarity.

Specific Feedback on Revisions

1. Quantitative Transparency in the Abstract The inclusion of specific means, standard deviations, and p-values in the Abstract is a major improvement. It now allows the reader to immediately grasp the empirical strength of the findings, particularly the notable enhancement of pain tolerance in the music and multimodal conditions. This level of detail is essential for a high-quality PLOS ONE publication.

2. Clarity of the Analytic Strategy I appreciate the reorganization of the Results section. The explicit use of headings for "Confirmatory Analyses" and "Exploratory Analyses" effectively guides the reader through the study’s logic and clearly distinguishes between pre-planned hypothesis testing and subsequent hypothesis-generating investigations. This approach greatly enhances the transparency of the work.

3. Justification for Manuscript Structure and Length I accept the authors' justification regarding the Stage 2 manuscript guidelines. Given that the Introduction and Methods must remain consistent with the approved Stage 1 protocol, the authors have done an excellent job of making minor adjustments to enhance readability without violating the Registered Report requirements.

4. Interpretation of Null Findings and Generalizability The revisions to the Discussion are well-noted. By acknowledging that the physiological and endocrine measures may have been underpowered, the authors provide a more scientifically grounded interpretation of the null results. Furthermore, the more cautious framing of the clinical implications within the context of the specific study sample (WEIRD, female-only) adds to the credibility of the conclusions.

A Minor Suggestion for the Final Version

Regarding the significant main effects reported in Table 2 for "Pain intensity" (p = .038) and "Pain affect" (p = .025), the authors correctly state that the post-hoc pairwise comparisons were not significant after Bonferroni correction. To ensure absolute clarity for future readers, I recommend that the authors conduct a final brief audit of the phrasing in these specific result paragraphs to ensure there is no ambiguity between the existence of an omnibus effect and the lack of specific group-level differences.

I am pleased to recommend this manuscript for publication and look forward to its contribution to the field of empirical aesthetics.

Reviewer #2: Thank you for the careful revision. The manuscript has improved, and the revised version is clearer in several important respects. In particular, the distinction between confirmatory and exploratory analyses is now more explicit, the abstract is more informative and quantitative, and the limitations regarding generalizability are better acknowledged.

I do have some comments that can be improved for the final version:

1. Exploratory analyses

The exploratory analyses are now more clearly identified and appear broadly reasonable. However, their justification and methodological strength are not fully convincing in all cases, particularly with regard to mechanistic interpretations.

2. Mechanistic interpretation

Some of the broader interpretive claims, especially those related to underlying mechanisms, still seem more tentative than the confirmatory results warrant. I encourage the authors to keep these interpretations cautious and clearly framed as exploratory.

3. Conclusions and scope of inference

The main conclusions regarding pain tolerance and the absence of consistent effects on stress markers are generally supported by the data and do address the core research questions. However, the broader implications should remain aligned with the strength of the findings.

4. Final framing

In the final version, it would be helpful to maintain a clear distinction throughout between supported confirmatory findings and more speculative or hypothesis-generating interpretations.

7. PLOS authors have the option to publish the peer review history of their article (what does this mean?). If published, this will include your full peer review and any attached files.). If published, this will include your full peer review and any attached files.

.

Reviewer #1: No

Reviewer #2: **Yes:**Itamar GrottoItamar Grotto

---

## [Author Response · Author response to Decision Letter 2]

23 Mar 2026

We have submitted a separate “Response to Reviewers” document that clarifies our responses in detail.

---

## [Editor Report · Decision Letter 2]

25 Mar 2026

Registered Report: How does Art Impact Pain and Stress?

Exposure to Multimodal Art (Music + Visual) and Music Alone Enhances Pain Tolerance More Than Visual Art, but Neither Art Form Impacts Autonomic or Endocrine Markers

PONE-D-25-45648R2

Dear Dr. Fekete,

We’re pleased to inform you that your manuscript has been judged scientifically suitable for publication and will be formally accepted for publication once it meets all outstanding technical requirements.

An invoice will be generated when your article is formally accepted. Please note, if your institution has a publishing partnership with PLOS and your article meets the relevant criteria, all or part of your publication costs will be covered. Please make sure your user information is up-to-date by logging into Editorial Manager at Editorial Manager® and clicking the ‘Update My Information' link at the top of the page. For questions related to billing, please contact  and clicking the ‘Update My Information' link at the top of the page. For questions related to billing, please contact billing support..

Kind regards,

Thiago P. Fernandes, PhD

Academic Editor

PLOS One
---

## [Editor Report · Acceptance letter]

PONE-D-25-45648R2

PLOS One

Dear Dr. Fekete,

I'm pleased to inform you that your manuscript has been deemed suitable for publication in PLOS One. Congratulations! Your manuscript is now being handed over to our production team.

Kind regards,

on behalf of

Dr. Thiago P. Fernandes

Academic Editor

PLOS One